# Characterizing the prevalence of *Fusobacterium necrophorum* subsp. *necrophorum, Fusobacterium necrophorum* subsp. *funduliforme,* and *Fusobacterium varium* in bovine and ovine semen, bovine gut, and vagino-uterine and fetal microbiota using targeted culturing and qPCR

Justine Kilama,[1] Carl R. Dahlen,[2] Mina Abbasi,[3] Xiaorong Shi,[3] T. G. Nagaraja,[3] Matthew S. Crouse,[4] Robert A. Cushman,[4] Alexandria P. Snider,[4] Kacie L. McCarthy,[5] Joel S. Caton,[2] Samat Amat[1]

**ABSTRACT** *Fusobacterium necrophorum* is an important pathogen associated with several infectious diseases in cattle. However, recent sequencing-based studies reported that *F. necrophorum* may be positively associated with pregnancy in beef cows and that *Fusobacterium* is highly abundant in bull seminal microbiota with potential involvement in reproductive health and fertility. Here, we performed a comprehensive screening to (i) determine the prevalence of *Fusobacterium necrophorum* (subspecies *necrophorum* [FNN] and *funduliforme* [FNF]) and *Fusobacterium varium* (FV) in the reproductive microbiota of cattle and sheep as well as bovine digestive tract ecosystems, and (ii) explore whether these *Fusobacterium* spp. colonize calf prenatally. For this, we screened 11 different sample types including bovine and ram semen, bovine vaginal and uterine swabs, and bull fecal samples, as well as samples from 180- and 260-day-old calf fetuses and their respective dams using both quantitative PCR (qPCR; 514 samples) and targeted culturing (499 samples). By qPCR, all the targeted *Fusobacterium* spp. were detected across all sample types with varying prevalence rates and viability. FNF was highly prevalent in the bull semen (66.7%) and maternal ruminal fluids (87.1%), and its viability was confirmed through culturing. All the targeted *Fusobacterium* spp. were identified in vaginal and uterine swab samples (3.1%–9.4%), caruncles, fetal fluids, rumen, and meconium samples (2.7%–26.3%) by qPCR but were not isolated by culture method. Overall, our results, for the first time, suggest that *F. necrophorum* is a commensal member of healthy male reproductive microbiota, and that FNF, FNN, and FV are present in bovine vagino-uterine microbiota and calf intestine prenatally.

**IMPORTANCE** Recent sequencing-based studies suggest that *Fusobacterium,* including *F. necrophorum*, a known primary etiological agent for several important infectious diseases in cattle, may be non-pathogenic members of the reproductive microbiota with pro-fertility effects. However, further information regarding the absolute abundance, viability, and higher taxonomic resolution of the *Fusobacterium* species and subspecies which cannot be achievable by the amplicon sequencing approach is needed to confirm the commensal and non-pathogenic status of the *Fusobacterium* spp. in cattle. Here, we performed a comprehensive screening of *F. necrophorum* subspecies *necrophorum*, *F. necrophorum* subspecies *funduliforme*, and *Fusobacterium varium* from over 500 samples from 11 different sample types using targeted culturing and qPCR. Overall, our results provide novel insights into the prevalence and viability of *Fusobacterium* spp. in bovine male and female reproductive tracts and their presence in calf fetuses, which will serve

**Peer Reviewer** Chenggang Wu, John P and Kathrine G McGovern Medical School at The University of Texas Health Science Center, Houston, Texas, USA

Address correspondence to Samat Amat, Samat.amat@ndsu.edu.

The authors declare no conflict of interest.

as the basis for further research into understanding the role of *Fusobacterium* in cattle fertility.

KEYWORDS *Fusobacterium* spp., male and female reproductive microbiome, fetus, cattle, qPCR, culture method, prevalence

*F*usobacterium necrophorum is a gram-negative, rod-shaped, non-spore-forming, non-motile, pleomorphic, and aerotolerant anaerobe that can colonize the gastrointestinal, respiratory, and reproductive tracts of both humans and animals (1–3). *F. necrophorum* is classified into two subspecies (subsp.), namely subsp. *necrophorum* (FNN) and subsp. *funduliforme* (FNF) (4). These subspecies exhibit distinct genetic (3), morphological, biochemical, and virulence characteristics (5), with FNN being notably more virulent than the FNF owing to its ability to produce more leukotoxin (6). Recently, *Fusobacterium varium* (FV), which shares characteristics with *F. necrophorum* in terms of habitat and biochemical niche such as indole production and the utilization of lactate and lysine (7–9), has been identified as the predominant *Fusobacterium* spp. in the bovine ruminal microbial ecosystem (10–12).

In addition to liver abscesses, necrotic laryngitis, and foot rot, *F. necrophorum* has been associated with mastitis and abortion in cattle (2, 13, 14). *Fusobacterium* spp. have also been implicated in the development of endometritis, especially in transition dairy cows experiencing metabolic stress and immunological impairments (15–17). Thus, *F. necrophorum* is considered as one of the most important bacterial pathogens implicated in many infectious diseases in both dairy and beef cattle (5, 13, 18). However, recent emerging evidence from culture-independent sequencing studies suggest that *Fusobacterium* spp., including *F. necrophorum,* might be a non-pathogenic commensal member of seminal (19) and vagino-uterine microbiota of healthy cattle (20). The phylum Fusobacteriota has been reported to account for 6.3% and 0.6% of the vaginal and uterine microbiota of healthy beef heifers, respectively (20), with noticeably high abundance of the genus *Fusobacterium* presenting in the vagina and uterus of these cattle (20). In addition, *F. necrophorum* was reported to be more abundant in the uterine microbiota of beef cows that became pregnant as compared to those that did not become pregnant (21). Together, these 16S rRNA gene amplicon sequencing-based studies highlight that the *Fusobacterium* spp., including *F. necrophorum,* may have a beneficial role in female cattle fertility.

Fusobacteriota is i reported to be a predominant phylum in the seminal microbiota of bovine bull (19, 22–25). Recently, we observed *Fusobacterium* as the most abundant genus accounting for 26% of total seminal microbiota in healthy and yearling beef bulls (19). We also reported a significant change and decline in the relative abundance of *Fusobacterium* in the bull semen during breeding season, possibly influenced by various factors (19). These factors may include potential transfer from the bull to the female reproductive tract (19) as well as other influences such as hormonal fluctuations, dietary changes, and physiological stress associated with breeding behaviors (26–28).

While the sequencing-based evidence highlights that *Fusobacterium* may be a commensal member of the bovine reproductive microbiota, the 16S rRNA gene amplicon sequencing approach does not provide information regarding the absolute abundance, cell viability, and higher taxonomic resolution of the *Fusobacterium* at both species and subspecies. Thus, to further confirm the findings from the previous sequencing-based studies, it is necessary to apply both culture-based and quantitative PCR (qPCR) methods to evaluate prevalence, concentrations, and viability of the *Fusobacterium* species and subspecies associated with the bovine reproductive microbiota. Given the potential variability in virulence and pathogenicity among different *Fusobacterium* species and subspecies, including *F. necrophorum*, it is essential to survey their prevalence within the bovine reproductive tract. In the present study, we employed both targeted culturing and qPCR methods to evaluate the prevalence, concentrations, and viability of FNN, FNF, and FV in 11 different sample types with

emphasis on reproductive tracts such as bovine vaginal and uterine samples as well as semen from bulls and rams. We hypothesize that the prevalence and viability of FNN, FNF, and FV differ across reproductive and digestive tract ecosystems in cattle and sheep. Additionally, given that there is growing evidence suggesting the existence of prenatal microbial colonization in calves (29–32),the secondary objective of this study was to explore the prevalence and viability of *Fusobacterium* spp. in samples associated with calf fetuses during late gestation. For this, we further hypothesize that *Fusobacterium* spp. may colonize the calf prenatally.

## MATERIALS AND METHODS

### Study designs, animal husbandry, dietary treatments and sample collection

To approach this research topic with a broad scope of sample types and animal species, we utilized samples from five different studies involving beef bulls, rams, beef heifers, and cows, as well as 180- and 260-day-old beef calf fetuses and their respective dams as illustrated in Fig. 1. While the diet and animal husbandry practices varied between different animal studies, sample collections and processing for microbial analysis from all animal studies were conducted following the same procedures by the same personnel.

The first study involved beef bulls with cohorts conducted in each of the 3 consecutive years, from 2022 to 2024. The detailed husbandry practices used for managing the bulls are described in our previous publication (19). In brief, 40 Meat Animal Research Center II bulls, aged 9–10 months old, were subjected to either moderate (1.13 kg/day) or high (1.80 kg/day) average daily weight gain feeding regimens over 112 days, with biweekly adjustments to feed delivery to ensure targeted body weight (BW) gain was achieved. The diet consisted of 25% alfalfa hay, 5% corn silage, 66% corn, and 4% vitamin/mineral pellets, administered via Calan gates (American Calan, Northwood, NH, USA). After being managed on pasture for 112 days, semen samples were collected from the bulls ($n$ = 36) at pre-breeding during the breeding soundness exams and then post-breeding, after a 28-day breeding season with 250 heifers (33). Briefly, the semen samples were collected through electroejaculation, using a collection handle lined with a plastic sheath (Pulsator IV; Lane Manufacturing Inc., Denver, CO, USA) as outlined in our previous publication (19). A 200 µL aliquot of the bull semen was transferred into cryotubes with 1 mL of 10% skim milk (Difco Skim Milk, Sparks, MD, USA) saturated with carbon dioxide gas ($CO_2$) and immediately placed on dry ice. Fecal samples were collected from the bulls through the rectum (~150 g) using plastic gloves (without additional lubrication) and placed into sterile Whirl-Pak bags. Approximately 5 mg of feces was then transferred into 1 mL of 10% skim milk that was saturated with $CO_2$. Both semen and fecal samples were frozen immediately and transported to the lab on dry ice for storage at −80°C until further analysis.

The second study involved rams, with a primary goal of evaluating the impacts of sire plane of nutrition on offspring outcomes. The study design with the husbandry practices and dietary management used the ram trial are described previously by (34). Briefly, 24 Rambouillet rams, aged 1.5 to 4 years, with an initial average BW of 82.9 ± 2.6 kg, were acclimated for 16 days at the NDSU Animal Nutrition and Physiology Center, where they were housed individually in temperature-controlled pens with simulated natural daylight. Over an 84-day feeding period, the rams were assigned to one of three nutritional planes: Positive, aiming for a 12% gain in BW; Maintenance, designed to maintain their current body weight; and Negative, targeting a 12% reduction in body weight. The ram labs (F1 generation) were fed a common diet for 7 months when semen samples were collected. Semen samples used in this study were collected from the mature rams (F0 generation) on days 0, 28, 56, and 84 using electroejaculation, with samples collected into a handle lined with a plastic sheath (34). A 50 µL portion of the semen was added to 1 mL of brain heart infusion (BHI) broth with 20% glycerol for culturing, and an additional 200 µL of the semen was stored in a 1.5 mL tube for qPCR assay following genomic DNA extraction.

**FIG 1** Illustration of experimental design, sample collection procedures, and screening techniques for *F. necrophorum* and *F. varium*.

The third study involved a herd of Angus-crossbred cows (27 nonpregnant, 31 pregnant) from which both uterine and vaginal swabs were collected at the time of artificial insemination (AI), as well as heifers (26 nonpregnant, 33 pregnant) from which only vaginal swabs were collected 2 days before AI to characterize the reproductive microbiota and its relation to pregnancy outcomes. A detailed description of the study design and the procedures for collection of both vaginal and uterine swabs are outlined in our previous publication (21). Briefly, the vulva was thoroughly cleaned with a sterile paper towel soaked in 70% ethanol. A sterile cotton-tipped applicator (15 cm, Puritan, Guilford, ME, USA) was then inserted into the midpoint of the vaginal cavity and swirled four times to ensure adequate contact with the vaginal wall and was withdrawn carefully to avoid contamination. The swab was immediately placed into a cryotube with 10% skim milk, stored on ice, and transported to the laboratory. Immediately after collecting the vaginal swabs, uterine swabs were obtained using a 71 cm double-guarded culture swab (Reproduction Provisions L.L.C., Walworth, WI, USA), which was carefully guided through the cervix into the uterine body with the aid of rectal palpation. The swab tip was extended through the inner plastic sheath, and gentle pressure was applied to the uterine body by pinching the swab and rotating it three times. The swab was then retracted into its sheaths and removed from the cow. The tip of the swab was cut off, placed into a 2 mL tube with 10% skim milk, kept on ice, and transported to the laboratory for processing.

The fourth study consisted of 20 beef heifers which were managed on either a high forage diet (75% forage, 25% concentrate) or a high-concentrate diet (75% concentrate, 25% forage) at the NDSU Beef Cattle Research Complex. The high forage diet comprised 54% alfalfa hay, 42% corn silage, and 4% vitamin/mineral pellet (dry matter basis), while the high-concentrate diet included 10% alfalfa hay, 30% corn silage, 56% corn, and 4% vitamin/mineral pellet (dry matter basis). The heifers were bred with AI using male-sexed semen and maintained on their respective dietary treatments until 180 days of gestation.

During gestation, ruminal fluid was collected from heifers using a pump-assisted rumen tube (20), aliquoted into cryotubes containing 10% skim milk, and immediately placed on dry ice before transportation to the laboratory for processing. Additionally, vaginal swabs were also collected as briefly outlined above and detailed in our previous publication (21). At day 180 of gestation, the cows were euthanised using captive bolt for fetal harvesting, which involved collection of samples from both the dams and fetuses. Samples collected from the dam consisted of ruminal fluids, vaginal swabs, and uterine swabs, while fetus-associated samples included allantoic and amniotic fluids, placental caruncle tissues, ruminal fluids, and meconium (Table 1). Immediately post-euthanasia, maternal ruminal fluid samples were collected after evisceration of the gastrointestinal tract. About 50 mL of ruminal fluid was drawn into a sterile 50 mL syringe (Medtronic, Minneapolis, MN, USA) using a sterile 22-gauge needle. The cranial vagina was swabbed as described above, and uterine swabs were obtained after exposing the gravid uterus from the dam during fetus harvesting. All samples were stored in 10% skim milk infused with $CO_2$ gas prior to immediate cryopreservation for subsequent DNA extraction and culturing.

As illustrated in Fig. 1, various fetus-associated samples, including allantoic and amniotic fluids, caruncles, ruminal fluids, and meconium, were collected under aseptic conditions. Allantoic and amniotic fluid samples were obtained as detailed in our previous publications (30, 35). Briefly, immediately after exposing the gravid uterus, up to 50 mL of allantoic and amniotic fluid was aspirated using a sterile 22-gauge needle into a sterile 50 mL syringe. Ruminal fluid samples from the fetus were collected after the rumen was cut open using a procedure outlined in our previous publication (30, 35). A 500 µL aliquot of each amniotic, allantoic, and rumen fluid sample was transferred into 1 mL of 10% skim milk infused with $CO_2$ gas and immediately placed on dry ice. Immediately after opening through the uterine walls and separation of caruncles from the fetal cotyledon, about 5.0 mg of maternal placental caruncle tissue was dissected (30), immersed in 1 mL of 10% skim milk infused with $CO_2$ gas, and placed on dry ice. Fetal meconium samples were collected into sterile Whirl-Pak bags, and a sterile spatula was used to transfer about 5.0 mg sample into 1 mL of 10% skim milk infused with carbon dioxide gas, followed by immediate freezing on dry ice.

The fifth study included a herd of 32 pregnant beef heifers, which were fed either control or restricted diets, with some receiving one-carbon metabolites (OCMs) supplementation, including methionine, choline, folate, and vitamin B12, as described previously (36). The heifers were inseminated with female-sexed semen and maintained on OCM dietary treatments until 260 days of gestation, at which time they were slaughtered and tissue from dams and fetuses was collected (S. Amat et al., unpublished

**TABLE 1** Sample size distribution of different types of samples screened for *F. necrophorum* and *F. varium* using qPCR and culturing methods

| Sample type | Number of samples screened | |
|---|---|---|
| | qPCR | Culturing |
| Bull semen | 96 | 116 |
| Bull feces | 24 | 24 |
| Bovine maternal ruminal fluid | 31 | 31 |
| Bovine vaginal swab | 55 | 68 |
| Bovine uterine swabs | 64 | 79 |
| Bovine caruncles | 38 | 38 |
| Bovine allantoic fluid | 34 | 34 |
| Bovine amniotic fluid | 37 | 37 |
| Fetal ruminal fluid | 31 | 31 |
| Fetal meconium | 31 | 31 |
| Ram semen | 100 | 10 |
| Total number of samples | 541 | 499 |

data). The procedures for sample collection from both the dam and fetuses post-euthanasia are consistent with those described in the previous studies (30, 35).

All the samples from the five studies were categorized into 11 sample types for analysis based on their respective sources: bull semen, bull feces, bovine maternal ruminal fluids, bovine vaginal swab, bovine uterine swabs, bovine caruncles, bovine allantoic fluids, bovine amniotic fluids, bovine fetal rumen fluids, bovine fetal meconium, and ram semen. The total number of each type of sample collected and utilized for either qPCR or targeted culturing are summarized in Table 1. Samples were preserved on dry ice and transported to the Department of Diagnostic Medicine and Pathobiology at Kansas State University (Manhattan, KS, USA) for qPCR analysis and targeted culturing method.

## Measures of contamination control

To control the risk of contamination during sample collection and processing, strict aseptic techniques and environmental controls were employed. Each sample was collected following sterile procedures to prevent cross-contamination. This included the use of single-use sterile gloves, syringes, needles, swabs, and collection tubes. Semen samples were collected aseptically, and the initial ejaculate fraction, which is more likely to contain contaminants from the urethra, was discarded. Additionally, we compared control swabs (prepuce, room air, and collecting sheath) with semen samples as a measure to assess potential contamination. For vaginal and uterine swabs, additional steps were taken, such as cleaning the vulva with paper towel soaked in 70% ethanol before sampling to reduce potential contamination from external sources. During the collection of maternal and fetal samples post-euthanasia, sterile instruments were used, and all handling was conducted aseptically to avoid introducing contaminants from the surrounding environment. Environmental controls were integrated throughout the collection process to monitor for potential contaminants. Sterile swabs were used to sample surgical trays, instruments, room air, and tap water from the collection area, providing baseline controls. These environmental samples were analyzed alongside the biological samples to identify any sources of contamination that might compromise the results. In addition, the $CO_2$ gas used to infuse the samples was passed through a 0.2 micron filter positioned midway along the tubing from the main gas tank, ensuring maximum filtration of potential contaminants from the $CO_2$ gas. To further reduce contamination risks, the infusion tip was carefully kept from contacting any samples, and it was thoroughly disinfected with 70% ethanol and replaced after each use in a few samples.

After collection, samples were immediately frozen on dry ice to maintain microbial integrity and transported to the laboratory, where they were stored at −80°C until further processing. While in the laboratory, rigorous aseptic protocols were followed to maintain sample integrity. Each sample type was processed in a sterile, designated work area to prevent cross-contamination. Separate tools and equipment were used for each type of sample, and processing took place in laminar flow hoods. All procedures were performed by trained personnel adhering to contamination control protocols. These stringent precautions safeguarded the sterility of each sample and preserved the integrity of the microbial community in each sample, ensuring that the results reflected true biological findings without cross-contamination and interference from external contaminants.

## Genomic DNA extraction

For the vaginal and uterine swab samples, DNA was extracted using procedures listed in our previous publication (21). For all other sample types, DNA extraction was conducted using the GeneClean Turbo Kit (MP Biomedicals, Solon, OH, USA) as previously described by (11). Briefly, 1 mL of each homogenized sample was boiled for 10 minutes and centrifuged at $9,300 \times g$ for 5 minutes. The supernatant (100 µL) was mixed with 500 µL of GeneClean Turbo Salt Solution and transferred to a cartridge, then centrifuged at $14,000 \times g$ for 5 seconds. After washing and additional centrifugation, DNA was eluted

**TABLE 2** Frequency of detection of targeted *Fusobacterium* spp. in different sample types using qPCR both before and after enrichment. The values represent the total number of samples that tested positive for each *Fusobacterium* species or subspecies across different sample types while the percentages in brackets indicate their prevalence relative to the total number of samples screened[a]

| Sample type | Number of samples | F. necrophorum subspecies funduliforme | | | F. necrophorum subspecies necrophorum | | | F. varium | | |
|---|---|---|---|---|---|---|---|---|---|---|
| | | Before enrichment | After enrichment | Total prevalence (%) | Before enrichment | After enrichment | Total prevalence (%) | Before enrichment | After enrichment | Total prevalence (%) |
| Bull semen | 96 | 25 (26.0%) | 39 (54.9%) | 64 (66.7%)* | 7 (7.3%) | 12 (13.5%) | 19 (19.8%)* | ND | 5 (5.2%) | 5 (5.2%)* |
| Bull feces | 24 | ND | ND | ND | ND | ND | ND | ND | 3 (12.5%) | 3 (12.5%)** |
| Bovine maternal ruminal fluids | 31 | 1 (3.2%) | 26 (86.7%) | 27 (87.1%)* | ND | 2 (6.5%) | 2 (6.5%)** | ND | ND | ND |
| Bovine vaginal swab | 55 | 1 (1.8%) | 2 (3.7%) | 3 (5.5%)** | ND | 2 (3.6%) | 2 (3.6%)** | 1 (1.8%) | 2 (3.7%) | 3 (5.5%)** |
| Bovine uterine swabs | 64 | ND | 2 (3.1%) | 2 (3.1%)** | 1 (1.6%) | 5 (7.9%) | 6 (9.4%)** | ND | 2 (3.1%) | 2 (3.1%)** |
| Bovine caruncles | 38 | ND | 8 (21.1%) | 8 (21.1%)** | 5 (13.2%) | 5 (15.2%) | 10 (26.3%)* | ND | 7 (18.4%) | 7 (18.4%)* |
| Bovine allantoic fluids | 34 | ND | 2 (5.9%) | 2 (5.9%)** | ND | 3 (8.8%) | 3 (8.8%)** | ND | ND | ND |
| Bovine amniotic fluids | 37 | ND | ND | ND | ND | 1 (2.7%) | 1 (2.7%)** | 4 (10.8%) | ND | 4 (10.8%)** |
| Fetal ruminal fluid | 31 | ND | 5 (16.1%) | 5 (16.1%)** | ND | 3 (9.7%) | 3 (9.7%)** | ND | 2 (6.4%) | 2 (6.4%)** |
| Fetal meconium | 31 | ND | ND | ND | ND | 2 (6.4%) | 2 (6.4%)** | ND | ND | ND |
| Ram semen | 100 | ND | ND | ND | 3 (3.0%) | ND | 3 (3.0%)*** | ND | ND | ND |
| Total | 541 | 27 (5.0%) | 84 (15.5%) | 111 (20.5%) | 16 (3.0%) | 35 (6.5%) | 51 (9.4%) | 5 (0.9%) | 21 (3.9%) | 26 (4.8%) |

[a]Different symbols (*, **, ***) within the same column of total prevalence represent a significant difference between sample types (P< 0.05) based on Fisher's exact test; ND, Not detected.

with 30 µL of GeneClean Turbo Elution Solution following a 5 minute incubation at room temperature and a final 1 minute centrifugation. The concentration and purity of extracted genomic DNA was analyzed using Nanodrop ND-1000 and followed by the Picogreen assay. The DNA was assessed for purity using a 260:230 ratio and deemed pure if the ratio ranged between 1.8 and 2.2 according to the Thermo Scientific NanoDrop Spectrophotometer protocol outlined in the technical bulletin-T042, with a concentration of at least 50 ng/µL. The eluted DNA was stored at −20°C until qPCR analysis.

## qPCR

A total of 541 genomic DNA samples, extracted from the 11 different sample types, before and after enrichment (described below), were analyzed using qPCR (Table 1) to detect and quantify the two subspecies of *Fusobacterium* and *F. varium* (11). The procedures for validation of the qPCR assay and determination of *Fusobacterium* gene copy numbers are outlined in the previous publication (11). Briefly, the *hgdA* gene, which encodes 2-hydroxyglutaryl dehydratase, was used for identification of *F. necrophorum* (*hgdA-n*) and *F. varium* (*hgdA-v*), and the promoter region (*lktA-n* and *lktA-f*) of the leukotoxin operon (*lktBAC*) was used to differentiate between the two subspecies of *F. necrophorum* (37). The assay protocol included an initial denaturation at 95°C for 5 minutes, followed by 45 amplification cycles consisting of 95°C for 15 seconds and 60°C for 40 seconds, performed in the BioRad CFX96 Real-Time System (BioRad, Hercules, CA). The targeted species or subspecies in different sample types were considered present when the qPCR assay was positive either pre- or post-enrichment (Table 2). Conversely, it was considered undetected if the qPCR assay yielded negative results both prior to and after enrichment. Notably, unlike conventional PCR, which primarily detects the presence or absence of a target sequence, qPCR not only identifies the presence of specific DNA but also quantifies the amount of target *Fusobacterium* DNA in the samples, offering both detection and precise quantification

## Targeted culturing

### Enrichment media

The culturing media used to enrich samples were made of pre-reduced, anaerobically sterilized peptone yeast extract medium (PY) with 100 mM lactate (PY-La) or lysine (PY-Ly) as the primary carbon source and supplemented with josamycin (3 µg/mL), vancomycin (4 µg/mL), and norfloxacin (1 µg/mL); PY-La JVN or PY-Ly JVN (11, 12) prior to isolation.

### Isolation of Fusobacterium spp

A total of 499 samples (Table 1) were subjected to targeted culturing both by direct plating without enrichment and plating after enrichment in PY-La JVN or PY-Ly JVN. A detailed protocol for isolation of *Fusobacterium* spp. has been previously described (12). In short, sample homogenates were spot inoculated onto blood agar (Remel Inc., Lenexa, KS), PY-La JVN agar, and PY-Ly JVN agar using sterile cotton swabs. Subsequently, an inoculating loop was used to streak from the inoculation spot to facilitate the isolation of single bacterial colonies. These inoculated plates were incubated in an anaerobic glove box at 37°C for 48 hours. Presumptive colonies, based on morphology shown in Fig. 3A (12), were selected and transferred onto blood agar plates and incubated anaerobically at 37°C for 48 hours. Confirmation of the species identification was performed by a qPCR assay targeting the *hgdA* gene. For enrichment, 1 mL of sample homogenate was inoculated into PY-La JVN and PY-Ly JVN broths, incubated at 37°C for 24 hours, and then streaked onto blood agar for isolation. The sample was considered positive for *Fusobacterium* spp. or subspecies if presumptive colonies exhibiting the morphological characteristics of the *Fusobacterium* species or subspecies were positive by qPCR.

## Statistical analysis

To assess the prevalence of *Fusobacterium* species and subspecies (FV, FNN, and FNF) across different sample types. The results of the qPCR assay and targeted culturing performed both before and after enrichment were individually combined to calculate the total prevalence (%) of FV, FNN, and FNF in each sample type. Using R software (version 4.4.0, "Puppy Cup," on a Windows platform, x86_64-w64-mingw32/x64), Fisher's exact test was performed to assess the association between the prevalence of *Fusobacterium* species or subspecies (FNF, FNN, and FV) across the sample types. For each sample type, a 2 × 3 contingency table was created to capture the binary outcome of prevalence (frequency of detected and undetected) of targeted *Fusobacterium* species or subspecies. The level of statistical significance was set at a *P*-value of <0.05. The cumulative prevalence data derived from each of the screening methods (either targeted culturing or qPCR) were displayed in figures created with Prism software (Prism 9.5.0, GraphPad).

## RESULTS

In the present study, we investigated the prevalence of FV, FNN, and FNF in the male and female reproductive tracts of cattle and sheep utilizing diverse sample types including vaginal and uterine samples from beef heifers and cows, as well as semen samples from beef bulls and rams (Fig. 1). Additionally, we evaluated the presence of *Fusobacterium* spp. in late-gestational calf fetuses and their respective dams to explore their prenatal colonization. Our findings showed significant differences in the prevalence of *Fusobacterium* species or subspecies across sample types (*P* < 0.05) as detected using both qPCR and targetted culturing method (Fig. 2).

## Detection of *Fusobacterium* spp. by qPCR

Of the 541 samples subjected to qPCR before and after enrichment, FNF was the most prevalent *Fusobacterium* subspecies in the majority of sample types analyzed (Fig. 3). The concentrations ranged from $1 \times 10^3$ to $1.5 \times 10^4$ CFU per milliliter, with an average concentration of $7 \times 10^3$ CFU per milliliter. The highest prevalence of FNF was in maternal ruminal fluid, with a prevalence of 87.1% followed by bull semen at 66.7%. FNF was also detected in 21.1% of bovine caruncles, and 16.1% of bovine fetal ruminal fluid, 5.9% of allantoic fluid, 5.5% of bovine maternal vaginal swabs, and 3.1% of uterine swabs. However, FNF was not detected in bull fecal samples, amniotic fluid, fetal meconium, and ram semen samples. Enrichment revealed the presence of FNF in bovine uterine swabs and fetal ruminal fluid (Table 2).

In contrast, FNN had a lower prevalence compared to FNF, although it was detected in almost all sample types examined with the exception of bull fecal samples. The highest prevalence of FNN was observed in bovine caruncles at 26.3%, followed by 19% in the bull semen. It was also present in 9.7% of fetal ruminal fluid, 9.4% of uterine swabs, 8.8% of allantoic fluid, and 6.5% of both maternal ruminal fluid and fetal meconium. Additionally, FNN was detected in 3.6% of maternal vaginal swabs, 3.0% of ram semen, and 2.7% of amniotic fluid.

Lastly, FV was the least prevalent of the *Fusobacterium* species across all the samples screened. It was detected most frequently in bovine caruncles (18.4%), followed by bull faeces (12.5%) and amniotic fluid (10.8%). FV was also found in 6.5% of fetal meconium, 5.5% of bovine maternal vaginal swabs, 5.2% of bull semen, and 3.1% of uterine swabs (Fig. 3). However, it was undetectable in bovine allantoic fluid, as well as in maternal and fetal ruminal fluid, and ram semen, even after enrichment.

## Culturing and isolation of *Fusobacterium* spp.

A total of 499 samples were subjected to targeted culturing to assess the viability and further confirm the prevalence results obtained from qPCR. For this, samples were

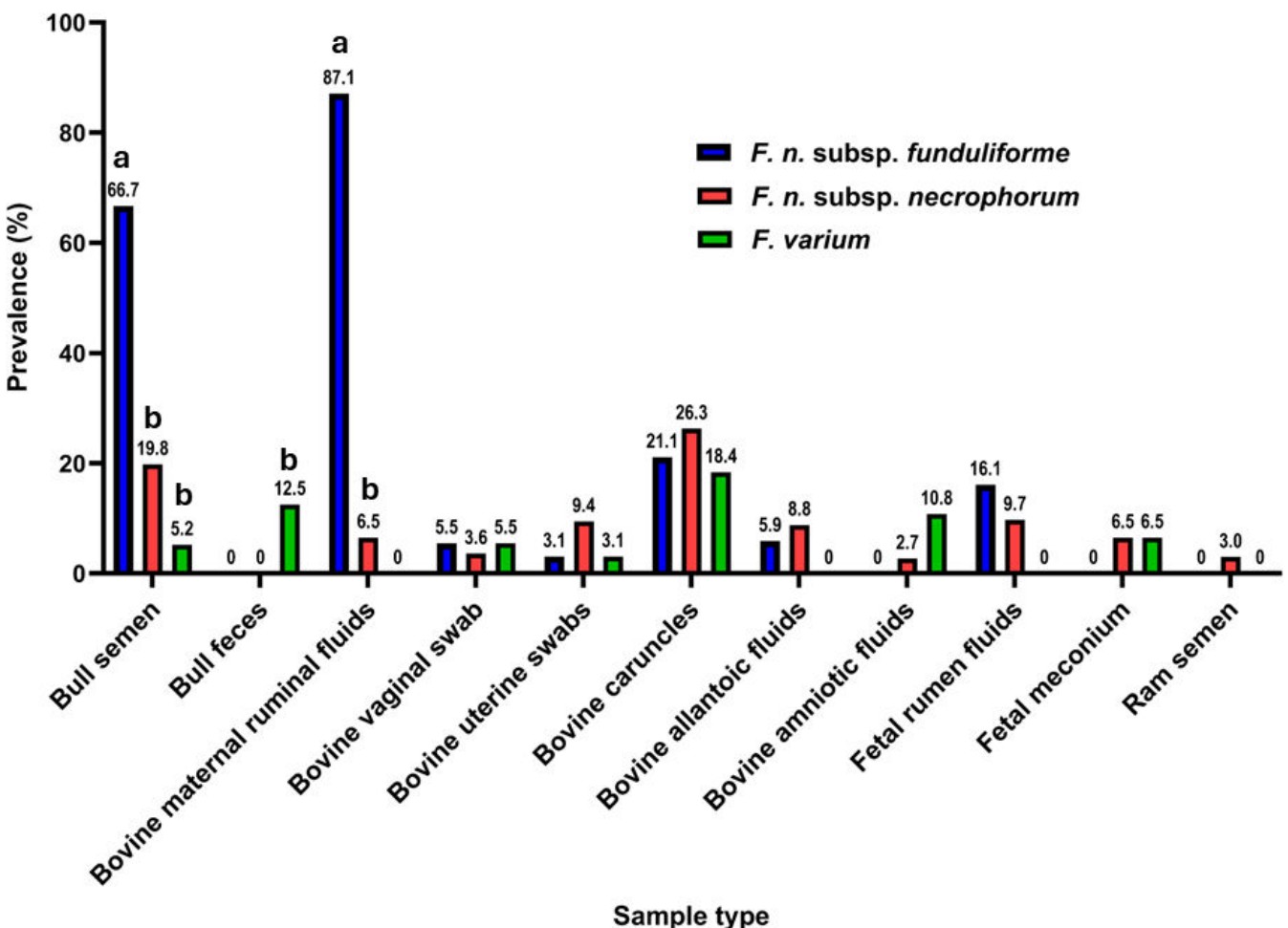

**FIG 3** Comparison of the total prevalence of FNF, FNN, and FV across different sample types ($n$ = 541) as screened using qPCR. The total prevalence (%) of FNF, FNN, and FV in each sample type was calculated by combining data from qPCR assay both pre- and post-enrichment. The data are presented as the proportion of samples positive for each *Fusobacterium* species or subspecies across the different sample types. Statistical analyses were performed using Fisher's exact test to assess the association between *Fusobacterium* prevalence across sample types. A $P$-value <0.05 was considered significantly different across each sample type (e.g., for sample types including bull semen and maternal ruminal fluids). The prevalence of *Fusobacterium* subspecies or species within each sample type with common letters is not significantly different.

cultured for the detection and isolation of FNN, FNF, and FV isolates by direct plating or plating after an enrichment step. Consistent with the qPCR results, FNF was the most frequently cultured subspecies. The greatest recovery of FNF was observed in the maternal ruminal fluid, where it was isolated from 67.7% of samples, followed by the bull semen at the prevalence rate of 62.9% (Fig. 4B). However, FNF was less frequently isolated from other sample types, with only a single isolate recovered from 68 bovine vaginal swabs, and none from uterine swabs, caruncles, allantoic fluids, amniotic fluids, fetal rumen fluid, fetal meconium, and ram semen (Fig. 4B). In bull semen, the number of isolates increased by 23 following enrichments, while maternal ruminal fluid and vaginal swabs yielded an additional 14 and 1 isolate, respectively (Table 3).

Only four isolates of FNN were obtained from bull semen through direct plating, with no isolates recovered following the enrichment step. Contrarily, a single FNN isolate was obtained from uterine swabs after enrichment. Surprisingly, FNN was not detected in all other sample types. On the other hand, FV was not isolated from any of the 499 samples analyzed, neither by direct plating nor after enrichment.

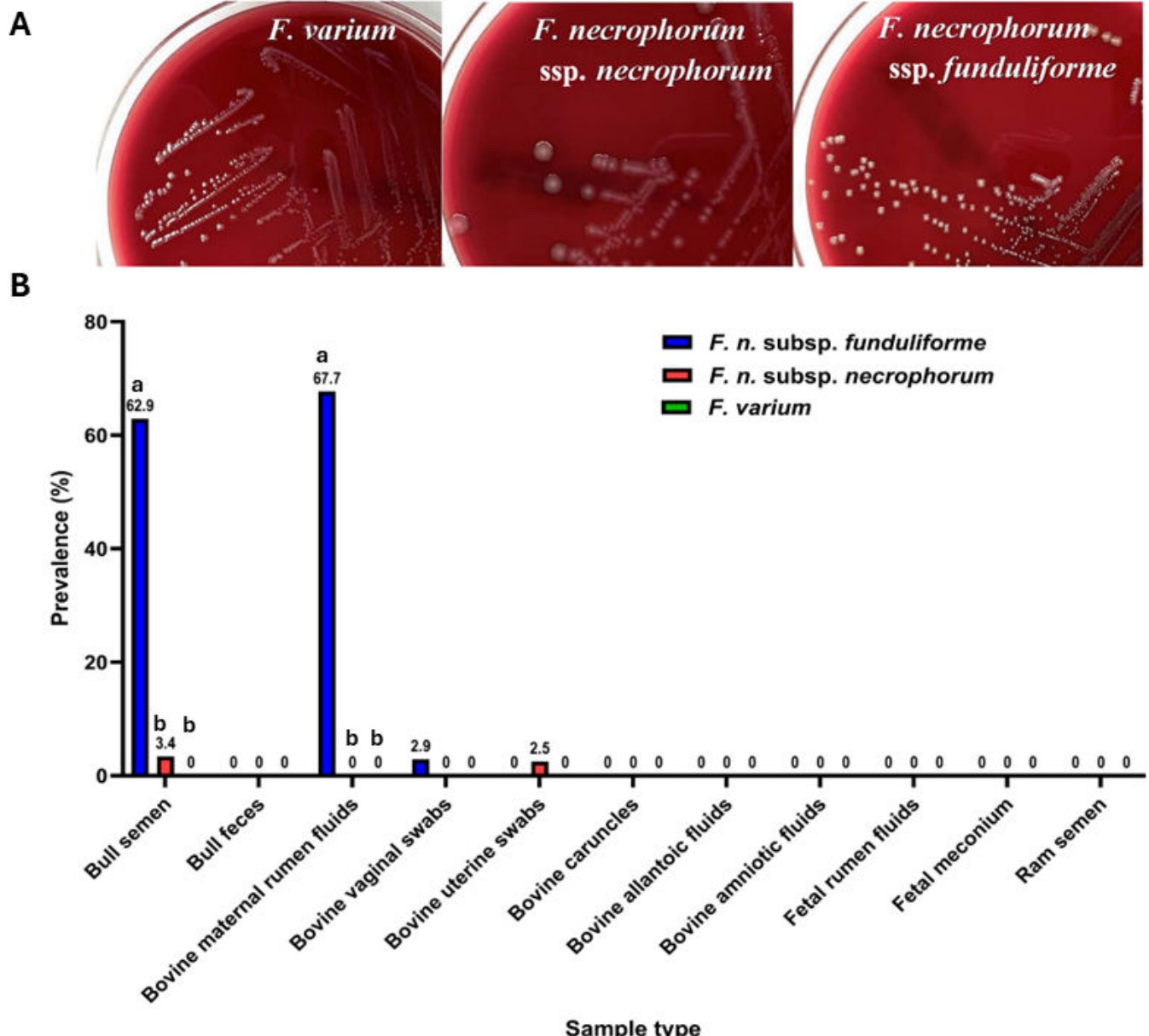

**FIG 4** Targeted culturing of *Fusobacterium* spp. (A) The colony characteristics of FV, FNN, and FNF isolated on blood agar media; adopted from Deters et al. (12); (B) prevalence and viability of FNF, FNN, and FV across different sample types (*n* = 499) as screened using targeted culturing techniques. The total prevalence (%) of FNF, FNN, and FV in each sample type was calculated by combining data from both culturing on blood agar both by direct plating and post-enrichment. The data are presented as the proportion of samples positive for each viable *Fusobacterium* species or subspecies across the different sample types. Statistical analyses were performed using Fisher's exact test to assess the association between *Fusobacterium* prevalence across sample types. A *P*-value <0.05 was considered significantly different across each sample type (e.g., for sample types including bull semen and maternal ruminal fluids). Within each sample type, subspecies or species with common letters are not significantly different.

## DISCUSSION

*Fusobacterium* species, particularly *F. necrophorum* and *F. varium,* that colonize the gastrointestinal, respiratory, and reproductive tracts of both animals and humans are often associated with infections (1–3, 13). With its long reputation as an opportunistic pathogen, *F. necrophorum* is implicated in a variety of infections, such as liver abscesses, foot rot, laryngitis, and metritis in cattle (6, 38–41) as well as Lemierre's syndrome in humans (42–44). Contrary to the traditional view of this bacterial species

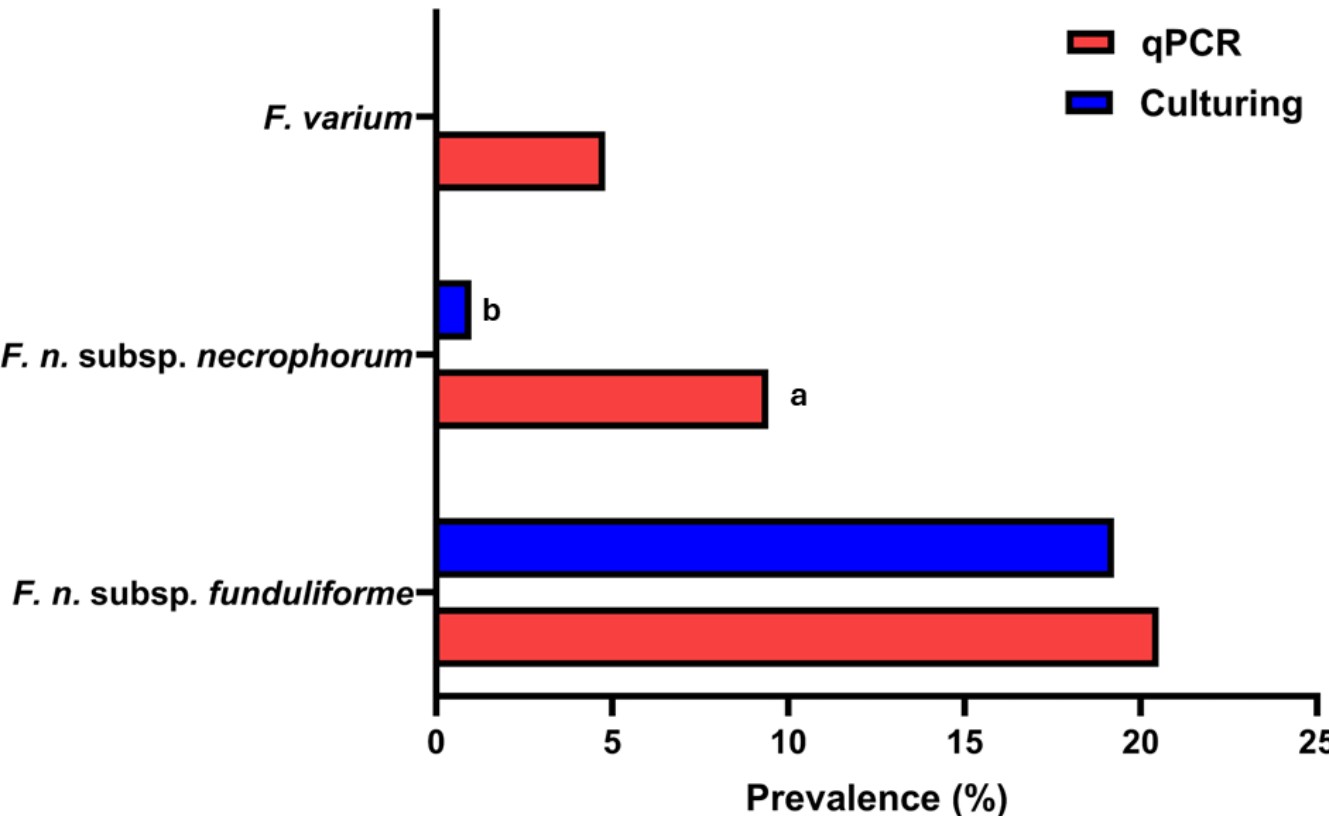

**FIG 2** Overall prevalence of targeted *Fusobacterium* spp. in all sample types screened using both qPCR (*n* = 541) and targeted culturing (*n* = 499). The total prevalence (%) of FNF, FNN, and FV in all screened samples was calculated by combining pre- and post-enrichment results from culture and qPCR assays, respectively. Enrichment means samples were cultured for 24 hours in a peptone yeast extract medium, utilizing either lactate (100 mM) or lysine (100 mM) as the main energy source, and supplemented with josamycin (3 µg/mL), vancomycin (4 µg/mL), and norfloxacin (1 µg/mL). Data are presented as the proportion of all samples positive for each *Fusobacterium* species or subspecies, either by qPCR or culturing. Fisher's exact test was employed to assess statistical differences in the proportions detected by qPCR and culturing. A *P*-value of less than 0.05 was used to determine significant differences within each species or subspecies. The *P*-values obtained were FNF (*P* = 0.856), FNN (*P* = 0.0185), and FV (*P* = 0.0594). Subspecies or species with common letters do not show significant differences in detection rates between qPCR and culturing methods.

being considered as merely pathogenic, in the present study, we presented evidence suggesting otherwise. We performed a comprehensive screening using both qPCR and targeted culturing to determine the prevalence of the *Fusobacterium* species, FNN, FNF, and FV in male (bovine and ovine semen) and female reproductive tracts, as well as microbial ecosystems associated with the rumen and bovine calf fetuses.

Using the qPCR assay, we observed that *Fusobacterium* spp. are prevalent in the semen of healthy yearling beef bulls at varying prevalence rates with FNF being the most abundant subspecies followed by FNN and FV. Through a combination of both direct plating and enrichment, we recovered the majority of the FNF isolates detected in bull semen using the qPCR assay. This confirms that there are viable FNF present in the reproductive tract of healthy male cattle and supports the findings from 16S rRNA gene amplicon sequencing studies reporting the high relative abundance of *Fusobacterium* in both semen (19, 24, 25) and prepuce of healthy bulls (23). In our previous 16S rRNA gene amplicon sequencing study, we identified *Fusobacterium* as the most relatively abundant bacterial genus, comprising an average of 26% of the total seminal microbiota sequencing reads in healthy beef bulls (19). Likewise, Medo and colleagues reported Fusobacteriota as the third most relatively abundant phylum, accounting for 18% of the bacterial microbiota present in semen samples of Slovak Holstein Friesian breeding bulls (24). The seminal microbiota of Slovak Holstein breeding bulls has been reported to be divided into two distinct clusters based on microbial composition, and

**TABLE 3** Frequency of isolation of targeted *Fusobacterium* spp. in different sample types both by direct plating and after enrichment: The values represent the total number of samples that tested positive for each *Fusobacterium* species or subspecies across different sample types while the percentages in brackets indicate their prevalence relative to the total number of samples screened[a]

| Sample type | Number of samples | F. necrophorum subspecies funduliforme | | | F. necrophorum subspecies necrophorum | | | F. varium | | |
|---|---|---|---|---|---|---|---|---|---|---|
| | | Isolation by direct plating | Isolation after enrichment | Total prevalence (%) | Isolation by direct plating | Isolation after enrichment | Total prevalence (%) | Isolation by direct plating | Isolation after enrichment | Total prevalence (%) |
| Bull semen | 116 | 50 (43.1%) | 23 (34.8%) | 73 (62.9%)* | 4 (3.4%) | ND | 4 (3.4%) | ND | ND | ND |
| Bull feces | 24 | ND | ND | ND | ND | ND | ND | ND | ND | ND |
| Bovine maternal ruminal fluids | 31 | 7 (22.6%) | 14 (58.3%) | 21 (67.7%)* | ND | ND | ND | ND | ND | ND |
| Bovine vaginal swab | 68 | 1 (1.5%) | 1 (1.5%) | 2 (2.9%)** | ND | ND | ND | ND | ND | ND |
| Bovine uterine swabs | 79 | ND | ND | ND | ND | 1 (1.3%) | 1 (1.3%) | ND | ND | ND |
| Bovine caruncles | 38 | ND | ND | ND | ND | ND | ND | ND | ND | ND |
| Bovine allantoic fluids | 34 | ND | ND | ND | ND | ND | ND | ND | ND | ND |
| Bovine amniotic fluids | 37 | ND | ND | ND | ND | ND | ND | ND | ND | ND |
| Fetal rumen fluid | 31 | ND | ND | ND | ND | ND | ND | ND | ND | ND |
| Fetal meconium | 31 | ND | ND | ND | ND | ND | ND | ND | ND | ND |
| Ram semen | 10 | ND | ND | ND | ND | ND | ND | ND | ND | ND |
| Total | 499 | 58 (11.6%) | 38 (7.6%) | 96 (19.2%) | 4 (0.8%) | 1 (0.2%) | 5 (1.0%) | 0 | 0 | 0 |

[a]Different symbols (*, **) within the same column of total prevalence represent a significant difference between sample types ($P < 0.05$) based on Fisher's exact test; ND, Not detected.

one of these clusters was dominated by the Fusobacteriota phylum (24). In addition, Koziol and colleagues reported that Fusobacteriota was one of the top seminal phyla in the healthy bull, along with Actinomycetota, Bacteroidota, Euryarchaeota, Bacillota, and Proteobacteria (19, 22). This phylum was also observed in the microbiota associated with bull prepuce, and *Fusobacterium* was one of the relatively most abundant genera in prepuce-associated microbiota (23). Based on our current qPCR and culturing results coupled with the existing 16S rRNA gene amplicon sequencing data (19, 24, 25) showing a high abundance of *Fusobacterium* in the bull semen, it is reasonable to speculate that *Fusobacterium* spp., particularly FNF, are part of the normal seminal microbial community. Thus, the presence of highly prevalent and viable FNF and moderately prevalent FNN in the healthy bull seminal microbiota calls for further research to understand the physiological function of *F. necrophorum* species in the male reproductive tract, their interactions with the seminal microbiota, and their involvement in reproductive immune system and sperm development, and bull fertility, as well as their influence on female cattle fertility.

In the present study, the notably high relative abundance of FNF in the bull semen compared to other screened *Fusobacterium* species may be attributed to its niche-specific adaptation and comparatively lower virulence. FNN is known to produce higher levels of leukotoxin (37, 45) as compared to FNF; therefore, less virulent FNF may make it better adapted for colonization and establishment in the bull reproductive microbial environment. As compared to bull semen, we observed a remarkably lower prevalence of *Fusobacterium* in ram semen. This finding highlights that the presence and abundance of FNF in the male reproductive tract may be animal host specific.

We recovered isolates of FNF from 62.9% (73/116) of the bull semen samples using combination of both direct culturing and enrichment techniques. Even though FNN and FV were prevalent in semen samples determined by qPCR, we only recovered a few FNN isolates and were not able to culture and isolate FV using our targeted culturing even after including an enrichment step. Our inability to culture FV from semen samples could be attributed to several factors. One possible factor could be that they require specific nutrients, growth factors, and culturing conditions (10, 46) which were not provided by the culturing medium and conditions that we used. Another factor might be associated with the lower number of viable FNN and FV cells present in the samples that we screened, which might be below the minimum detection limit. Another explanation could be associated with the genomic and metabolic characteristics of FNN and FV that made it challenging to culture. Thus, metagenomic sequencing of the bull seminal microbiome and metagenome-assembled genomes (MAGs) of seminal FNN and FV should be performed by future studies as the MAGs can help to identify key metabolic pathways and nutrient requirements encoded within the genes, which will then allow determining the nutrient requirements and growth factors (47).

Although FNF, FNN, and FV were highly prevalent in the bull seminal samples, we observed significantly lower detection of these *Fusobacterium* spp. in female reproductive tract, with a prevalence ranging between 3.1% and 9.8% in vaginal and uterine samples. These qPCR results are not consistent with the reports derived from the 16S rRNA gene amplicon sequencing-based studies. We previously reported a relatively high abundance of *Fusobacterium* in the uterus of healthy beef heifers (20), and that *F. necrophorum* was differentially more abundant in the uterine microbiota of cows that successfully conceived than those that remained open after AI with log2 fold change of 5.36 (21). *Fusobacterium* has been reported to be a common member of the reproductive microbiota in healthy female cattle (20, 21, 48). *Fusobacterium* was also identified as a key genus in both the vaginal and cervical microbiota of healthy dairy heifers and cows (49–51). The relatively low prevalence of *Fusobacterium* spp. detected in vaginal and uterine swabs by qPCR in our current study compared to those studies relied on 16S rRNA sequencing can be attributed to many factors. The 16S rRNA gene sequencing (V3-V4 regions or V4 regions) provides data on relative abundance of *Fusobacterium* at the genus level, and this genus encompasses 13 different species (52, 53). The qPCR

method used in the present study, however, specifically targeted amplification of DNAs from one *Fusobacterium* species (*F. varium* or *F. necrophorum*). Thus, this partially explains the discrepancy between the results from the 16S rRNA gene sequencing-based studies and our present study. Second possibility is that the screened FNF, FNN, and FV might be present at low abundance in the vaginal and uterine microenvironments whose abundance is under the minimum detection limit of the qPCR method.

As compared to prevalence determined by qPCR, relatively small number of FNF (2.9% in vaginal swabs) and FNN (2.5% in uterine swabs) isolates were recovered via culturing. These culturing results indicate that there are viable FNF and FNN cells present in the upper and lower reproductive tract of healthy female cattle. Our previous 16S rRNA sequencing-based studies revealed that the *F. necrophorum* was enriched in the uterus of cows that became pregnant to AI at the time of breeding as compared to that of cows that failed to become pregnant . No FV isolates were recovered from any of the vaginal and uterine swab samples screened, suggesting that this *Fusobacterium* species may not be culturable by the growth media and culturing conditions applied in the present study, or they may present as non-viable or vegetative state.

While the comparative genomic analysis is yet to be done on the isolate genomes or MAGs of FNF, FNN, and FV originating from the male and female reproductive tracts, the detection of these *Fusobacterium* spp. in the seminal, vaginal, and uterine microbial ecosystems of healthy cattle suggests that *F. necrophorum* may transfer between the male and female reproductive tracts. Microbes associated with semen are believed to "hitchhike" to the uterus and can influence the microbial landscape of the female reproductive tract (54). A statistically strong similarity was observed between men's seminal and women's vaginal microbiota after sexual intercourse, suggesting the presence of the complementary semino-vaginal microbiome in human couples (55). We recently observed a noticeable decline in the relative abundance of seminal *Fusobacterium* during breeding in beef bulls, with a 7% decline in its abundance over the 28 days of breeding season (19). While the diet, age, and environment (pasture vs confinement housing) could have contributed to this reduction in *Fusobacterium* abundance during breeding, we also speculate on the possibility of transfer of *Fusobacterium* from the bull semen to the urogenital tract of heifers during mating (54). A number of studies also emphasized the dynamic interplay between the seminal and vaginal microbiomes, proposing that the seminal bacteria could integrate into the vaginal environment, creating a temporary but influential shift in microbial composition, which may influence uterine health, either contributing to dysbiosis or promoting a balanced microbiome (56, 57). Accordingly, we proposed that the introduction of seminal *Fusobacterium* spp. into the female reproductive system during mating could affect not only fertility and the development of the embryo, but it may also transfer from the female to the offspring calf perinatally.

In the present study, we identified a significantly high prevalence of FNN in various fetus-associated samples, including caruncles (maternal portion of the placenta) (26.3%), fetal ruminal fluid (9.7%), allantoic fluid (8.8%), maternal ruminal fluid (6.5%), meconium (6.5%), and amniotic fluid (2.7%). Moreover, FNF was also detected in caruncles (21%), allantoic fluid, and fetal ruminal fluid samples (5.9% to 21%). In addition, caruncles, amniotic fluid, and meconium samples were also positive for FV. Collectively, these qPCR data suggest that the two *F. necrophorum* subspecies may colonize calves prenatally. While the concept of *in utero* microbial colonization still remains controversial, with many still adhering to the "sterile-womb hypothesis," which suggests that microbiome acquisition occurs only during and after birth (58, 59), recent studies (60–63) done in humans have provided convincing evidence supporting the former hypothesis. Until very recently, it was believed that colonization of rumen by various microbes, including methanogenic archaea, starts only at or after birth (64, 65). However, this belief has been challenged by recent evidence showing that microbial colonization of the bovine fetal intestine may take place *in utero* (30–32, 66). For example, Guzman et al. (32) reported the presence of microbiota in five different components of the gastrointestinal tract and

amniotic fluid obtained from 5, 6, and 7 months of gestation calf fetuses (mid gestation) in a study using both molecular- and culture-based approaches along with stringent contamination controls. This report was further supported by our own study which suggested that colonization of fetal intestine by bacterial community may take place within the first 12 weeks of gestation in cattle (30). In addition, 16S rRNA gene amplicon sequencing-based evaluation of the amniotic and meconium samples obtained from full-term calves delivered via Caesarean section also revealed the existence of amniotic and meconium microbiota (66). Therefore, identification of *Fusobacterium* spp. in samples associated with 260-day-old calf fetuses in the present study also contributes to the growing body of evidence supporting the presence of *in utero* microbial colonization in cattle.

None of the FNN, FNF, or FV was cultured from any of the fetal-associated samples screened. This finding could indicate that the *Fusobacterium* spp. present in calf prenatally might be in a non-viable state. Despite being not viable, the presence of *F. necrophorum* cells *in utero* and fetal intestine may have implications for fetal immune and microbial programming. Recently, it has been reported that both innate and adaptive immune systems are present with established T and B cell receptor diversity as early as 4 months of gestational age, and maturation of the human intestinal immune system starts early in fetal development (67). Additionally, studies suggest that bacteria present in the fetal intestine and amniotic fluid, along with maternal antibodies capable of retaining microbial molecules and transferring them to the offspring during pregnancy, may play a crucial role in imprinting the fetal immune system (60, 68). For instance, the bacterial extracellular vesicles in amniotic fluid were reported to resemble those from the maternal gut microbiota and involved in priming the fetal immune system for postnatal gut colonization (69). Moreover, bacterial strains (e.g., *Staphylococcus* and *Lactobacillus* spp.) found in fetal tissues such as gut, skin, lung, thymus, spleen, and placenta have been reported to stimulate memory T cells in mesenteric lymph nodes of the fetus *in vitro* (63). It has also been reported that metabolites transferred from the maternal microbiota to the fetus are important for the development of fetal innate immune cells, while the postnatal microbiota further supports the maturation of immune tissues and cells (70–72). Therefore, our current detection of FNN, FNF, and FV in both maternal uterine, placenta, and fetuses suggests that *Fusobacterium* species may be commensal members of the bovine reproductive tract that colonize calf prenatally. Our study poses an important research question: whether prenatal colonization of *F. necrophorum* has any impact on the postnatal susceptibility of a calf to *F. necrophorum*-associated infectious diseases such as liver abscesses, mastitis, endometritis, and foot rot in cattle.

Overall, despite the high detection rates of *Fusobacterium* spp. across various samples using qPCR, only 53.7% (101/188) of the detected microbes were successfully cultured and isolated by direct plating and enrichment. The majority of these cultured isolates were FNF from bull semen and bovine ruminal fluids, with none recovered from the fetal-associated samples. The generally lower culturing and isolation rates of FNF, FNN, and FV compared to the qPCR detection rates could be due to several factors. It is possible that many *Fusobacterium* spp., particularly those detected in fetus-associated samples using qPCR, may be present in non-viable form, or they require special growth media and culturing conditions. Another factor potentially contributing to the discrepancy observed between qPCR detection and culturing results could be reduction in *Fusobacterium* viability because of exposure to aerobic conditions during sampling (73). *Fusobacterium* spp. are obligatory anaerobes and, thus, exposure to oxygen during sample collection can significantly diminish their viability (74). To minimize the oxygen exposure during samples, we stored the samples in pre-reduced 10% skim milk infused with $CO_2$. Skim milk is regarded as an effective cryopreservation medium, aiding in the preservation of bacterial stocks thawed from −80°C (75). While efforts were made to plate the samples as fresh as possible, the geographical distance between the sampling sites and the microbiology laboratory necessitated freezing prior to plating. Hence, sample freezing and thawing may have further impacted the viability of some

*Fusobacterium* spp. in the screened samples (76). Consequently, the culturing results may underrepresent the viable *Fusobacterium* species, particularly those susceptible to freeze-thaw cycles and oxygen exposure.

The current study has both strengths and limitations. We employed a relatively robust methodology, combining both culture-based and qPCR techniques, and provided complementary information to the 16S rRNA amplicon sequencing. Additionally, using relatively large samples comprised of 11 different sample types and derived from five distinct trials ensured more robust and comprehensive insights into the prevalence of *Fusobacterium* spp. in both male and female bovine reproductive tracts and fetal samples as well as ovine semen. Furthermore, we applied uniform procedures of microbial sample collection, followed strict aseptic protocols to minimize potential contaminations, and ensured consistent processing of all samples. The authors acknowledge the following limitations associated with this study: (i) culturing *Fusobacterium* spp. might be compromised due to the potential aerobic exposure during sampling and processing, and using the same culturing media and culture conditions for all different sample types; (ii) our data is largely limited to the prevalence of *Fusobacterium* spp. and lack data showing the genomic and phenotypic characteristics of FNF, FNN, and FV isolates isolated from seminal, vaginal, uterine, ruminal, and fecal samples. Despite these limitations, our study, for the first time, provides novel and comprehensive information on the prevalence of *Fusobacterium* spp., which are important bacteria that have implications for several economically important infectious diseases in cattle, bull semen, female reproductive tract, and calf fetuses. This information serves as an important basis in guiding future studies to explore the potential role that *Fusobacterium* spp. may have in bull and cow fertility, and its possible involvement in calf prenatal immune programming and postnatal disease resilience. Additionally, our results also highlight that *Fusobacterium* spp. could be used as a model bacterial organism for studying the microbial transfer from bull-to-heifer-to-calf, as well as the role of microbiota in transferring maternal and paternal programming effects to the offspring.

## Conclusions

Among the *Fusobacterium* spp. screened, FNF was the most prevalent subspecies, particularly in bull semen and bovine maternal ruminal fluids. The second most frequently identified *Fusobacterium* spp. across screened different samples were FNN. All three *Fusobacterium* spp. were present in bull semen, vaginal, and uterine samples. FNN was present in all fetal-associated samples with greater prevalence as compared to FNF and FV. The overall prevalence rate of all *Fusobacterium* spp. as determined by culturing was significantly lower than that of qPCR. Although enrichment steps generally improved isolation rate for FNF, they did not significantly enhance the culturing recovery of FNN or FV. Overall, our results suggest that *F. necrophorum* is potentially a commensal member of healthy bovine male reproductive microbiota, and that FNF, FNN, and FV are present in bovine vagino-uterine microbiota and calf intestine prenatally. Our findings highlight the need for further research to investigate the potential roles of *F. necrophorum* and other *Fusobacterium* species in cattle fertility, their potential transmission from bull to heifer to calf, as well as *in utero* fetal immune programming. Additionally, further investigation into the potential involvement of *Fusobacterium* species in both paternal and maternal programming of offspring is warranted.

## ACKNOWLEDGMENTS

This study was funded by the North Dakota Agricultural Experiment Station as part of a start-up package for S.A. Mention of a trade name, proprietary product, or specific agreement does not constitute a guarantee or warranty by the USDA and does not imply approval to the inclusion of other products that may be suitable. USDA is an equal opportunity provider and employer.

J.K. and S.A. conceptualized the study, designed the sampling strategy, and drafted the initial manuscript. S.A., J.K., C.R.D., M.S.C., R.A.C., A.P.S., K.L.M., J.S.C., and S.A. were responsible for sample collection. M.A., X.S., and T.G.N. handled sample processing and analyses. All authors contributed to the manuscript writing, revision, editing, and finalization and approved the submitted version.

## AUTHOR AFFILIATIONS

[1]Department of Microbiological Sciences, North Dakota State University, Fargo, North Dakota, USA

[2]Department of Animal Sciences, Center for Nutrition and Pregnancy, North Dakota State University, Fargo, North Dakota, USA

[3]Department of Diagnostic Medicine and Pathobiology, Kansas State University, Manhattan, Kansas, USA

[4]USDA, Agriculture Research Service, U.S. Meat Animal Research Center, Clay Center, Nebraska, USA

[5]Department of Animal Sciences, Institute of Agriculture and Natural Resources, University of Nebraska-Lincoln, Lincoln, Nebraska, USA

## AUTHOR ORCIDs

Justine Kilama  http://orcid.org/0009-0002-8375-8994
Samat Amat  http://orcid.org/0000-0002-6824-2431

## AUTHOR CONTRIBUTIONS

Justine Kilama, Conceptualization, Data curation, Formal analysis, Investigation, Methodology, Visualization, Writing – original draft, Writing – review and editing | Carl R. Dahlen, Investigation, Methodology, Writing – review and editing | Mina Abbasi, Data curation, Investigation, Methodology, Validation, Writing – review and editing | Xiaorong Shi, Data curation, Investigation, Methodology, Validation, Writing – review and editing | T. G. Nagaraja, Conceptualization, Data curation, Formal analysis, Investigation, Methodology, Supervision, Validation, Visualization, Writing – review and editing | Matthew S. Crouse, Investigation, Methodology, Writing – review and editing | Robert A. Cushman, Investigation, Methodology, Writing – review and editing | Alexandria P. Snider, Investigation, Methodology, Writing – review and editing | Kacie L. McCarthy, Investigation, Methodology, Writing – review and editing | Joel S. Caton, Investigation, Methodology, Writing – review and editing | Samat Amat, Conceptualization, Data curation, Formal analysis, Funding acquisition, Investigation, Methodology, Supervision, Validation, Visualization, Writing – original draft, Writing – review and editing

## DATA AVAILABILITY

All the data are presented in the manuscript in figure and table format.

## ETHICS APPROVAL

All the animal handling and experimental procedures in this study were approved by the relevant institutional committees for animal research. The protocols involving bulls were approved by the Institutional Animal Care and Use Committee (IACUC) for the U.S. Meat Animal Research Center (USMARC IACUC; experiment #147.2). The protocols authorized by the IACUC at North Dakota State University (NDSU) include cows and heifers study (#A21061), 180-day-old fetuses and dams (IACUC20210043), and 260-day-old fetuses (#A21049).

## ADDITIONAL FILES

The following material is available online.

## Open Peer Review

**PEER REVIEW HISTORY (review-history.pdf).** An accounting of the reviewer comments and feedback.

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
