## [Reviewer comments · Microbiology Spectrum]

Microbiology Spectrum

Characterizing the prevalence of *Fusobacterium necrophorum* subsp. *necrophorum*, *Fusobacterium necrophorum* subsp. *funduliforme* and *Fusobacterium varium* in bovine and ovine semen, bovine gut, vagino-uterine and fetal microbiota using targeted culturing and qPCR

Justine Kilama, Carl Dahlen, Mina Abbasi, Xiaorong Shi, T. Nagaraja, Matthew Crouse, Robert Cushman, Alexandria Snider, Kacie McCarthy, Joel Caton, and Samat Amat

Corresponding Author(s): Samat Amat, North Dakota State University

Review Timeline:

Submission Date:	December 2, 2024
Editorial Decision:	December 19, 2024
Revision Received:	January 26, 2025
Accepted:	January 31, 2025

Editor: Artem Rogovsky

Reviewer(s): Disclosure of reviewer identity is with reference to reviewer comments included in decision letter(s). The following individuals involved in review of your submission have agreed to reveal their identity: Chenggang Wu (Reviewer #2)

Transaction Report:

DOI: <https://doi.org/10.1128/spectrum.03145-24>

Re: Spectrum03145-24 (Characterizing the prevalence *Fusobacterium necrophorum* and *Fusobacterium varium* in bovine and ovine semen, bovine gut, vagino-uterine and fetal microbiota using targeted culturing and qPCR)

Dear Dr. Samat Amat:

Thank you for the privilege of reviewing your work. Below you will find my comments, instructions from the Spectrum editorial office, and the reviewer comments.

Revision Guidelines

Sincerely,
Artem Rogovsky
Editor
Microbiology Spectrum

Reviewer #1 (Comments for the Author):

This is a well-written paper with intriguing results. Minor revisions are noted in the attached document, which should be addressed prior to publication. Some key recommendations include:

Hypothesis Alignment: Revise the hypothesis to align with the data and methods presented in the study. The current hypothesis

is not addressed or supported by the research as designed and conducted.

Ram Data Removal: Remove the ram data from the manuscript. The collection methods and analyses for these samples differ from those used for the bovine samples, and the negligible presence of *Fusobacterium* spp. in the ram data does not contribute to the paper's main findings. Excluding this data will strengthen the overall narrative.

Statistical Analysis Expansion: Provide a more thorough explanation of the statistical methods, particularly concerning how treatment effects were considered in the studies included in this analysis. If treatment effects were not accounted for, include a justification for this decision.

Enhanced Figure Captions: Revise and expand figure captions to ensure that each figure can stand alone. Captions should include details about statistical analyses, sample types, and any significance indicated in the figures.

Reviewer #2 (Comments for the Author):

Kilama et al., in this manuscript, report the prevalence of *Fusobacterium necrophorum* (FNN, FNF) and *Fusobacterium varium* (FV) in reproductive microbiota of cattle and sheep, as well as in digestive tract ecosystems, using PCR and culture methods. Their findings reveal a high prevalence of FNF in bull semen (66.7%) and maternal ruminal fluids (87.1%). Importantly, their study is the first to suggest that *F. necrophorum* may be a commensal member of the healthy male reproductive microbiota. Overall, this manuscript is well-written, with a robust experimental design and sound implementation. The research provides novel insights into the existence and distribution of *F. necrophorum* in cattle and sheep. The unexpected discovery of FNF being more prevalent than FNN and FV in cattle semen and ruminal fluids challenges previous findings that suggested FNN was the dominant species. The authors' explanation of these discrepancies is reasonable and highlights the complexity of studying *F. necrophorum*'s presence and pathogenicity.

I only have two minor comments

1) P87-90: The sentences in this section are somewhat repetitive and could be revised for better clarity and conciseness. For instance:

"We and our group have previously reported that..."

2) The rationale for using qRT-PCR instead of conventional PCR to differentiate target genes for FNN, FNF, and FV requires clarification. Typically, qRT-PCR is employed to quantify gene expression levels, while conventional PCR, using specific primers, is more commonly used to identify specific strains based on the presence or size of the PCR product. Although qRT-PCR can potentially achieve similar results, it may not be as straightforward or practical for this purpose. Please elaborate on the decision to use qRT-PCR over conventional PCR. Additionally, providing a detailed description of the PCR protocols and primers would improve the manuscript's transparency and reproducibility.

Reviewer #3 (Comments for the Author):

Summary: The manuscript from Kilama, et al describes a large-scale survey of the prevalence and viability of *Fusobacterium necrophorum* subsp. *necrophorum* and subsp. *funduliforme*, as well as *Fusobacterium varium* within bovine and ovine reproductive tracts. Some *Fusobacterium* species are traditionally thought of as livestock pathogens but previous work from this group suggests that some may also play a role in supporting fertility. Samples of bull and ram semen, bovine vaginal swabs and uterine tissue, as well as amniotic fluid and prenatal calf gut were screened via quantitative PCR for the presence of *Fusobacterium* using primer sets that distinguished between subsp. *necrophorum*, subsp. *funduliforme*, and *F. varium*. All three *Fusobacteria* were detected via this method, with bull semen and bovine maternal ruminal fluid having high prevalence of subsp. *funduliforme*. Paired samples were enriched for growth and plated onto media to confirm the presence of viable bacteria, with subsp. *funduliforme* being the overwhelming majority of detected species, again primarily in the bull semen and bovine maternal ruminal fluid. Species identity was confirmed via qPCR. Overall, this is a straightforward manuscript with assays that are well-performed, and the observations that *Fusobacterium necrophorum* subsp. *necrophorum* may be linked to fertility is therefore of economic interest for those in the livestock industry.

Specific comments:

1) How do the authors discriminate true seminal bacteria from those of the external bull urogenital tract? The semen was collected post-ejaculation, so it would have contacted any bacteria along the urethra. The authors even note that *Fusobacteria* are present in the bull prepuce in a previous study, so are the results in the current manuscript detecting seminal bacteria or urethral/prepuce bacteria? This seems to be an important distinction, even if the end result is the same, i.e., ejaculated semen from most bulls has subsp. *funduliforme* in it.

2) Do the authors have enough data to detect fertility differences between bulls and dams with viable *Fusobacteria* vs. those that were only detected via qPCR?

3) The discussion section is the longest part of the manuscript by quite a margin and is somewhat repetitive. Removing

redundancy and narrowing the focus would strengthen this section and improve readability.

4) The text in the blue bar of table 3 has some words randomly between columns, so it is hard to judge which column they are supposed to be linked to.

Spectrum 03145-24 Review

Line 3: Why not include *Fusobacterium funduliforme* in the title or change to *Fusobacterium* spp. I understand that the *funduliforme* is a sub-species but it can be confusing to readers not familiar with microbial taxonomy.

Line 55: *Fusobacterium** (spelling)

Line 82: Add space after “cattle”

Line 90-92: The microbiome could change for multiple reasons during the breeding season- not just due to transfer to the female. This is addressed somewhat in the discussion but some statements regarding this concept are reaching with the current data presented.

Line 107-9: Change the hypothesis to match this study- the study designs and current data presented does not answer or address this hypothesis. This could be a hypothesis in follow up research with an experimental design to address this question specifically.

Line 114-22: This section should be edited and shortened for ease of reading.

“protocols authorized by the IUCAC at NDSU include cows and heifers (#A21061), 180 day fetuses and dams (IUCAC20210043), and 260 day old fetuses (#A21049)”

Line 139: Ram data needs to be removed from this study. The results from the samples do not add to the story authors are trying to tell and sample collection methods were different.

Line 171: Combine the paragraphs from the study design section and sample collection section so there is a clear understanding of how specific animals were sampled and samples were processed. Remove ram data. It may also be good to refer to what these samples are being named in the graphs and data presentation of this paper. I commend authors on combining these data sets but more thorough explanation of what samples (data presented) came from which study should be provided.

Line 256: Not all samples?? Also it may be better to mention this earlier in the materials and methods as some readers could be confused about the preservation of samples in 10% skim milk and CO₂.

Line 238: Thank you for including this section, it is good for readers to reference for standards to prevent sample contamination.

Line 268: What samples and how many of each sample type underwent DNA extraction for qPCR?

Line 270: How were swabs processed- states homogenized samples but not sure if swabs underwent a different extraction process. Add a statement

Line 277: What was the Nanodrop purity/concentration cut offs for DNA?

Line 314: Did authors include any of the trt from the individual studies in the model first prior to grouping the samples- while the overarching goal of this study is to determine the presence of *Fusobacterium* spp. if bulls on high planes of nutrition have higher concentrations that is a finding worth mentioning to the readers. If this was analyzed in previous research articles still briefly mention and cite in results before moving into data for this study to give reader a better overarching picture.

Line 391: Yearling* bulls

Line 448: List percentages for the uterine microbiota to define “significantly more abundant” in this statement.

Line 479: Define “relatively strong” in context of statistics?

Line 483: If there was no sampling of females it is hard to say a decrease in bull reproduct abundance directly correlates to an increase in the female.

Line 509: “Guzman et al.”

Line 524: “Recently” spelling

Line 525: remove space between “both innate”

Line 529-32: This citation/sentence seems out of place here. Not tied in well. Remove or rewrite.

Line 533-34: citation?

Line 539: “shaping the fetal immune system” this is reaching for the data presented in this study.

Line 585: Again, there was no data presented in this study showing evidence of viable *Fusobacterium* spp in the prenatal calf, so these conclusions are reaching.

Line 599-600: healthy bovine* male reproductive tract

Line 605: Add “is warranted” or some iteration at the end of the sentence.

Figures:

Figure 1:

Remove ram data

Add *F. funduliforme* to the figure captions.

Shorten qPCR description: “Primers were designed to target the *hgdA* gene and probes for the leukotoxin promoter regions were used to identify *F. necrophorum* (*lktA-n*), *F.n.fundeuliforme* (*lktA-f*), and *F. varium* (*hgdA-v* gene).”

Figure 2:

Not sure why significance is only denoted in bull semen and bovine maternal ruminal fluids. Is this analysis within sample type? How was this significance determined? Figures and figure captions need to stand alone so more description regarding the statistical differences and analysis for all figures is needed.

Remove ram data

Figure 3:

Remove ram data

Again, figure captions and figures should stand alone. What was “Adopted from 12”?

Figure 4:

Remove ram data from total prevalence data.

“*Fusobacterium* spp in ALL*(?) sample types”

Figure needs to stand alone- type of graph and analysis that created this graph.
Statistical significance??

Table 1:

Remove ram data

Indent all lines over 1 to create space between “bovine maternal ruminal fluid” and the qPCR number

Table 2:

Remove ram semen

Would aid reader if you color blocked the table by *Fusobacterium* spp.

Need to move the Table 2 caption to be below the table.

$P < 0.05^*$

Table 3:

Column names are not correctly formatted

Need to move Table 3 caption below the table

$P < 0.05^*$

Overall this was a well written paper with relevant information regarding *Fusobacterium* presence in the reproductive tract of livestock. Authors should remove the ram data as it does not fit well into the overall story of the paper. There was some reaching in the discussion about programming and immune development potential of *Fusobacterium* that was not supported by the data in this study. Be aware of those types of statements and correct as needed.

Spectrum 03145-24 Review

Line 3: Consider including *Fusobacterium funduliforme* in the title or changing it to *Fusobacterium spp.* While *funduliforme* is a subspecies, it could confuse readers unfamiliar with microbial taxonomy.

Line 55: Correct the spelling of *Fusobacterium*.

Line 82: Add a space after “cattle.”

Lines 90-92: The microbiome could change for multiple reasons during the breeding season, not solely due to transfer to the female. Although this is addressed somewhat in the discussion, some statements are overstated given the current data.

Lines 107-109: Revise the hypothesis to align with the study. The current data and study design do not address the hypothesis presented. This could be reframed as a potential hypothesis for follow-up research with a specific experimental design.

Lines 114-122: Edit and condense this section for readability:
“Protocols authorized by the IUCAC at NDSU include cows and heifers (#A21061), 180-day fetuses and dams (#IUCAC20210043), and 260-day-old fetuses (#A21049).”

Line 139: Remove ram data. The results from these samples do not contribute to the narrative and differ in collection methods.

Line 171: Combine the paragraphs in the study design and sample collection sections for clarity. Remove ram data. Clearly explain how specific animals were sampled and how samples were processed prior to freezing. Refer to how these samples are named in graphs and data presentation. While merging datasets is commendable, provide a more detailed explanation of which samples came from which study.

Line 256: Clarify whether all samples were preserved in this manner. This detail might be better placed earlier in the materials and methods to prevent confusion about preservation methods (e.g., 10% skim milk and CO₂).

Line 238: Including this section on standards to prevent sample contamination is appreciated. It will be helpful for readers.

Line 268: Specify which samples and how many underwent DNA extraction for qPCR.

Line 270: Explain how swabs were processed. If swabs underwent a different extraction process than homogenized samples, add a statement.

Line 277: Include Nanodrop purity/concentration cutoffs for DNA.

Line 314: Did authors include treatments from individual studies in the model before grouping samples? If bulls on high planes of nutrition had higher *Fusobacterium spp.* concentrations, this is noteworthy. Mention previous research results briefly if applicable and cite them before presenting this study's findings.

Line 391: Correct "yearling* bulls."

Line 448: Define "significantly more abundant" with percentages for the uterine microbiota.

Line 479: Clarify the term "relatively strong" in a statistical context.

Line 483: Without female sampling, it is speculative to state that decreased bull reproductive tract abundance correlates with increased female abundance.

Line 509: Correct citation to "Guzman et al."

Line 524: Correct the spelling of "Recently."

Line 525: Remove the space between "both innate."

Lines 529-532: This citation and sentence are misplaced. Rewrite or remove for coherence.

Lines 533-534: Add a citation.

Line 539: The claim "shaping the fetal immune system" overextends the study's data.

Line 585: The study does not provide evidence of viable *Fusobacterium spp.* in prenatal calves. Avoid unsupported conclusions.

Lines 599-600: Use "healthy bovine* male reproductive tract."

Line 605: Add "is warranted" to the sentence end.

Figures:

Figure 1:

- Remove ram data.
- Add *F. funduliforme* to figure captions.
- Shorten qPCR description to:
“Primers targeted the *hgdA* gene, and probes for the leukotoxin promoter regions identified *F. necrophorum* (lktA-n), *F. n. fundeuliforme* (lktA-f), and *F. varium* (*hgdA-v* gene).”

Figure 2:

- Explain why significance is only noted in bull semen and maternal ruminal fluids. Was the analysis within sample type? Include statistical methods in captions.
- Remove ram data.

Figure 3:

- Remove ram data.
- Ensure figure captions explain terms like “Adopted from 12.”

Figure 4:

- Remove ram data from total prevalence data.
- Clarify “*Fusobacterium* spp in ALL*(?) sample types.”
- Specify graph type, analysis methods, and significance.

Tables:

• Table 1:

- Remove ram data.
- Add indentation to create spacing between “bovine maternal ruminal fluid” and the qPCR number.

• Table 2:

- Remove ram semen.
- Color-block by *Fusobacterium* spp. for clarity.
- Move the caption below the table.
- Use “ $P < 0.05^*$.”

• Table 3:

- Correct column formatting.
- Move the caption below the table.
- Use “ $P < 0.05^*$.”

Summary:

The paper is well-written and provides relevant insights into *Fusobacterium* in livestock reproductive tracts. Removing ram data would improve focus. Address overreaching conclusions in the discussion, particularly regarding programming and immune development, which are not supported by the data.

RESPONSE TO EDITOR AND REVIEWERS:

Manuscript # Spectrum03145-24

Full Title: Characterizing the prevalence of *Fusobacterium necrophorum* and *Fusobacterium varium* in bovine and ovine semen, bovine gut, vagino-uterine and fetal microbiota using targeted culturing and qPCR

RESPONSE: We thank the associate editor and reviewers for their thorough evaluation and providing insightful input on our manuscript. We have carefully addressed all feedback and made significant revisions, including enhancing scientific rigor, clarifying methodologies, aligning the hypothesis with our findings, and improving the focus and clarity of our discussion section. Specific comments on statistical methods, data presentation, and formatting have also been addressed as suggested. Our revised manuscript "Marked-up Manuscript," with revisions shown using Track Changes attached for your review. We look forward to your evaluation of this improved version. We have considered and addressed all comments raised by the 3 reviewers point by point as provided below:

REVIEWER COMMENTS:

Reviewer #1 (Comments for the Author):

This is a well-written paper with intriguing results. Minor revisions are noted in the attached document, which should be addressed prior to publication. Some key recommendations include:

Response: We thank the Reviewer for the positive comments on our manuscript. We have addressed the minor revisions highlighted and made the necessary adjustments accordingly as shown below.

Hypothesis Alignment: Revise the hypothesis to align with the data and methods presented in the study. The current hypothesis is not addressed or supported by the research as designed and conducted.

Response: As suggested, we have revised the hypothesis to better align with the data and methods outlined in the study (Lines 110-120).

Ram Data Removal: Remove the ram data from the manuscript. The collection methods and analyses for these samples differ from those used for the bovine samples, and the negligible presence of *Fusobacterium spp.* in the ram data does not contribute to the paper's main findings. Excluding this data will strengthen the overall narrative.

Response: Thank you for your positive feedback on our manuscript. We greatly appreciate your valuable suggestions, which have significantly improved the revised version. Regarding your suggestion to remove the ram semen data, we think that including the ram semen data in the manuscript enriches the overall narrative by providing a comparative perspective across bovine

and ovine animal species. Although methodologies for collection differed between rams and bovines, the data generated serve as an important counterpoint, demonstrating how variations in microbiome composition may be influenced by host species-specific factors. The authors would like to provide the following justifications for retaining this data:

1). Including the ram semen data enriches the manuscript by providing a comparative perspective on *Fusobacterium* prevalence between bovine and ovine species. While the ram semen data is not the central focus of this manuscript, it still offers valuable insights into host species-specific differences. For instance, the observation that *Fusobacterium* spp. is more prevalent in bull semen than in ram semen could guide future research, including comparative studies of microbiome dynamics across species.

2). Although the prevalence of *Fusobacterium* spp. in ram semen is low, these findings highlight potential biological differences and contribute to a broader understanding of microbiome-host interactions. Removing the data would limit the scope and diminish the broader significance of the study.

3). Additionally, significant resources, time, and effort were invested in collecting ram semen samples, performing DNA extractions, and running qPCR assays. Excluding this data would mean wasting these substantial efforts and resources, which we hope to avoid.

4). Furthermore, removing the ram semen data would reduce the total sample size screened for *Fusobacterium* using qPCR by 100 samples, thereby narrowing the study's scope and reducing its impact. It would also require reanalysis of all data, revisions to figures, and substantial rewriting of the manuscript.

5). Finally, neither the Editor nor the other two reviewers raised concerns or suggested removing the ram semen data. Their feedback did not indicate that including this data compromises the quality or focus of the manuscript.

For these reasons, we wish to retain the ram semen data in the manuscript. We believe, its inclusion expands the study's scope and provides a foundation for future research into host species-specific microbiome differences. We kindly request the Reviewer's understanding and hope they will agree to retain the ram semen data in the manuscript.

Statistical Analysis Expansion: Provide a more thorough explanation of the statistical methods, particularly concerning how treatment effects were considered in the studies included in this analysis. If treatment effects were not accounted for, include a justification for this decision.

Response: We thank the Reviewer for this comment. The primary focus of this study was on the prevalence of *Fusobacterium* spp. across different sample types, rather than assessing the impact of treatment on its prevalence and viability. The samples were derived from various studies with distinct treatment designs, and as such, the analysis was specifically tailored to explore *Fusobacterium* presence as a commensal in these sample types, irrespective of treatment. This approach allowed us to highlight broad ecological patterns across the sample types.

In future studies, we plan to design trials that specifically investigate the effects of treatment, not only on the prevalence of *Fusobacterium*, but also its mechanistic roles in fertility and immunomodulation. This will provide a more targeted approach to understanding the dynamic relationship between *Fusobacterium* and host factors under controlled treatment conditions.

Enhanced Figure Captions: Revise and expand figure captions to ensure that each figure can stand alone. Captions should include details about statistical analyses, sample types, and any significance indicated in the figures.

Response: We thank the reviewer for this suggestion. We have revised and expanded the figure captions as recommended. We added more details on statistical analyses, sample types, and significance. This revision made each figure can stand alone and provides a clearer understanding of the results (Lines 719-767).

Line 3: Why not include *Fusobacterium funduliforme* in the title or change to *Fusobacterium* spp. I understand that the *funduliforme* is a sub-species but it can be confusing to readers not familiar with microbial taxonomy.

Response: As suggested, we have updated the title to encompass the two subspecies of *Fusobacterium necrophorum* to precisely represent all species and subspecies included in the study.

Line 55: *Fusobacterium** (spelling)

Response: Corrected.

Line 82: Add space after “cattle”

Response: Changed as suggested.

Line 90-92: The microbiome could change for multiple reasons during the breeding season- not just due to transfer to the female. This is addressed somewhat in the discussion but some statements regarding this concept are reaching with the current data presented.

Response: We thank the Reviewer for pointing out this. We have revised the statement to acknowledge other potential factors that may have contributed to these changes, such as hormonal fluctuations, dietary changes, and physiological stress during breeding, with the relevant references cited (Lines 88-97).

Line 107-9: Change the hypothesis to match this study- the study designs and current data presented does not answer or address this hypothesis. This could be a hypothesis in follow up research with an experimental design to address this question specifically.

Response: We have revised the hypothesis to align with the study design and data presented (Lines 110-120).

Line 114-22: This section should be edited and shortened for ease of reading. “protocols authorized by the IUCAC at NDSU include cows and heifers (#A21061), 180 day fetuses and dams (IUCAC20210043), and 260 day old fetues (#A21049)”

Response: As suggested, we edited and revised the statements (Line 127-134).

Line 139: Ram data needs to be removed from this study. The results from the samples do not add to the story authors are trying to tell and sample collection methods were different.

Response: As addressed above, we would like to keep the ram semen data. The low prevalence of *Fusobacterium* in ram semen could suggest that *Fusobacterium* spp. might be animal host specific. This finding, regardless of collection methods, we believe, still be valuable for guiding future studies which maybe be necessary to validate these findings and expand the investigation to other livestock species.

Line 171: Combine the paragraphs from the study design section and sample collection section so there is a clear understanding of how specific animals were sampled, and samples were processed. Remove ram data. It may also be good to refer to what these samples are being named in the graphs and data presentation of this paper. I commend authors on combining these data sets, but more thorough explanation of what samples (data presented) came from which study should be provided.

Response: We have combined the study design and sample collection sections for clarity (Lines 135,149-162, 173-178, 184-201, 208-236, 242-309, 318-322, 336-338). A statement clarifying the grouping of sample types from different studies has also been added. However, we would like to retain the ram semen data as it provides valuable insights into species-specific prevalence of *Fusobacterium* spp., information that could guide future research endeavors.

Line 256: Not all samples?? Also it may be better to mention this earlier in the materials and methods as some readers could be confused about the preservation of samples in 10% skim milk and CO₂.

Response: The statement was revised as suggested (Lines 306-308).

Line 238: Thank you for including this section, it is good for readers to reference for standards to prevent sample contamination.

Response: We thank the Reviewer for this positive comment.

Line 268: What samples and how many of each sample type underwent DNA extraction for qPCR?

Response: The information about number of each sample types is provided in Table 1. Sample size distribution of different types of samples screened for *F. necrophorum* and *F. varium* using qPCR and culturing methods.

Line 270: How were swabs processed- states homogenized samples but not sure if swabs underwent a different extraction process. Add a statement

Response: Statement added as the DNA from the swabs were extracted using a different method listed in our previous publication.

- Webb EM, Holman DB, Schmidt KN, Pun B, Sedivec KK, Hurlbert JL, Bochantin KA, Ward AK, Dahlen CR, Amat S. Sequencing and culture-based characterization of the

vaginal and uterine microbiota in beef cattle that became pregnant or remained open following artificial insemination. *Microbiol Spectr.* 2023 Dec 12;11(6):e0273223. doi: 10.1128/spectrum.02732-23. Epub 2023 Nov 3. PMID: 37921486; PMCID: PMC10714821.

Line 277: What was the Nanodrop purity/concentration cut offs for DNA?

Response: The DNA was assessed for purity using a 260/230 ratio and deemed pure if the ratio fell between 1.8–2.2, with a concentration of at least 50 ng/μL. A statement was added to clarify this in the revised manuscript. Information regarding the DNA purity assessment was derived from https://dna.uga.edu/wp-content/uploads/sites/51/2019/02/Note-on-the-260_280-and-260_230-Ratios.pdf

Line 314: Did authors include any of the trt from the individual studies in the model first prior to grouping the samples- while the overarching goal of this study is to determine the presence of *Fusobacterium* spp. if bulls on high planes of nutrition have higher concentrations that is a finding worth mentioning to the readers. If this was analyzed in previous research articles still briefly mention and cite in results before moving into data for this study to give reader a better overarching picture.

Response: This study focused on screening for the presence and viability of *Fusobacterium* spp., leveraging samples from multiple studies to enhance its scope. The effects of individual treatments were not analyzed, as this would require a dedicated study specifically designed to evaluate treatment impacts under controlled conditions. The primary objectives of the original studies differed from this paper and were reported elsewhere. Since this study focuses on screening *Fusobacterium* spp. presence and viability, including treatment effects from individual studies would be beyond its scope.

Line 391: Yearling* bulls

Response: Corrected

Line 448: List percentages for the uterine microbiota to define “significantly more abundant” in this statement.

Response: The differential abundance of ASVs classified as *F. necrophorum* was analyzed using MaAsLin, which evaluates log₂ fold changes rather than percentages. A positive log₂ fold change indicates higher abundance in pregnant cows, with *F. necrophorum* showing a significant log₂ fold change of 5.36. This clarification has been added to the manuscript.

Line 479: Define “relatively strong” in context of statistics?

Response: Yes, Mandar and colleagues reported a strong and statistically significant concordance between semen and vaginal microbiome data, with $m12 = 0.870$ before intercourse, $m12 = 0.867$

after intercourse, and $m12 = 0.843$ based on changes in OTU relative abundances, all with p-values < 0.001 .

- Mändar R, Punab M, Borovkova N, Lapp E, Kiiker R, Korrovits P, Metspalu A, Krjutškov K, Nõlvak H, Preem JK, Oopkaup K, Salumets A, Truu J. Complementary seminovaginal microbiome in couples. *Res Microbiol.* 2015 Jun;166(5):440-447. doi: 10.1016/j.resmic.2015.03.009. Epub 2015 Apr 11. PMID: 25869222.

Line 483: If there was no sampling of females it is hard to say a decrease in bull repro tract abundance directly correlates to an increase in the female.

Response: As you have rightfully pointed out, further studies involving simultaneous sampling from both bulls and cows, before and after breeding, would be essential to establish a clear correlation. We have stated that this concept is speculative, based on our current understanding of the potential interchange of microbiota between males and females from the human study conducted by Mandar and colleagues (cited above).

Line 509: “Guzman et al.”

Response: Corrected as suggested.

Line 524: “Recently” spelling

Response: Corrected as suggested.

Line 525: remove space between “both innate”

Response: Removed as suggested.

Line 529-32: This citation/sentence seems out of place here. Not tied in well. Remove or rewrite.

Response: The sentence has been revised to enhance clarity and better alignment with the context.

Line 533-34: citation?

Response: Relevant citation has been added.

Line 539: “shaping the fetal immune system” this is reaching for the data presented in this study.

Response: We revised the statement (Lines 631-632).

Line 585: Again, there was no data presented in this study showing evidence of viable *Fusobacterium* spp in the prenatal calf, so these conclusions are reaching.

Response: While this study does not present direct evidence of viable *Fusobacterium* spp. in prenatal calves, the detection via qPCR provides a foundation for future research to explore the

potential role of *Fusobacterium* spp. in bull and cow fertility, as well as its influence on calf prenatal immune programming and postnatal disease resilience.

Line 599-600: healthy bovine* male reproductive tract

Response: Corrected as suggested

Line 605: Add “is warranted” or some iteration at the end of the sentence.

Response: Added as suggested.

Figures:

Figure 1:

Remove ram data

Response: As stated previously, the authors wish to retain the ram semen data because much as it has low prevalence, the data still provides valuable insights into host species-specific prevalence of *Fusobacterium* spp., being more prevalent in bull semen but not ram semen. This information could guide future research endeavors.

Add *F. funduliforme* to the figure captions.

Response: Added as suggested.

Shorten qPCR description: “Primers were designed to target the hgdA gene and probes for the leukotoxin promoter regions were used to identify *F. necrophorum* (lktA-n), *F.n.funduliforme* (lktA-f), and *F. varium* (hgdA-v gene).”

Response: The qPCR description shortened as suggested (Figure 1).

Figure 2:

Not sure why significance is only denoted in bull semen and bovine maternal ruminal fluids. Is this analysis within sample type? How was this significance determined? Figures and figure captions need to stand alone so more description regarding the statistical differences and analysis for all figures is needed.

Response: Captions have been revised to elaborate the statistical analyses for all figures and it can be interpretable as a stand alone.

Remove ram data

Response: As mentioned above, we would like to keep the ram semen data because much as it has low prevalence, the data still provides valuable insights into host species-specific prevalence of *Fusobacterium* spp., being more prevalent in bull semen but not ram semen. This information could guide future research endeavors including comparative studies.

Figure 3:

- Remove ram data:

Response: Addressed above.

- Again, figure captions and figures should stand alone. What was “Adopted from 12”?

Response: The footnote has been revised to ensure the figure can be interpreted as a stand alone. The phrase "Adapted from 12" was included to acknowledge that the figure originates from our previous publication. Deters A, Shi X, Ty L, Nagaraja1 PTG. 2024. First report of isolation of *Fusobacterium varium* from liver abscesses and ruminal and colonic epithelial tissues of feedlot cattle. *Applied Animal Science* 40:244–249.

Figure 4:

- Remove ram data from total prevalence data.

Response: Addressed above.

- “Fusobacterium spp in ALL*(?) sample types”

Response: Corrected as suggested.

- Figure needs to stand alone- type of graph and analysis that created this graph. Statistical significance??

Response: We have revised the footnote and added relevant statistical significance to make the figure a stand alone.

Table 1:

- Remove ram data

Response: Addressed above.

- Indent all lines over 1 to create space between “bovine maternal ruminal fluid” and the qPCR number

- **Response:** Corrected as suggested.

Table 2:

- Remove ram semen

Response: Addressed above.

- Would aid reader if you color blocked the table by *Fusobacterium* spp.

Response: Color blocking added.

- Need to move the Table 2 caption to be below the table.

Response: The table title is placed above, and the footnote is included below the table.

- $P < 0.05^*$

Response: Corrected as suggested.

Table 3:

- Column names are not correctly formatted
Response: the formatting of column names has been corrected.
- Need to move Table 3 caption below the table
Response: The same as table 2.
 $P < 0.05^*$
Response: Corrected as suggested

Summary:

Overall this was a well written paper with relevant information regarding *Fusobacterium* presence in the reproductive tract of livestock. Authors should remove the ram data as it does not fit well into the overall story of the paper. There was some reaching in the discussion about programming and immune development potential of *Fusobacterium* that was not supported by the data in this study. Be aware of those types of statements and correct as needed.

Response: Thank you for your positive feedback on our manuscript. We thank you once again for providing valuable feedback and detailed suggestions, which helped improving revised manuscript significantly. With respect your suggestion on removing the ram semen data from the manuscript, as mentioned above, we would like to keep the ram semen data. Although prevalence *Fusobacterium* in ram semen samples was low, it still offers valuable insights into the host species-specific prevalence of *Fusobacterium* spp. Although the ram semen data isn't the main story in this manuscript, this data still shows *Fusobacterium* is more prevalent in bull semen than in ram semen, which could guide future research, including comparative studies. In addition, we put into significant resources in ram semen sample collection, DNA extraction, and running qPCR. Including ram semen samples in this study expands the scope and improves the significance of the present study. Finally, neither Editor nor the two other Reviewers who reviewed this manuscript didn't suggest removal of the ram semen data. Therefore, we would like to keep the ram semen data and would very much appreciate if the Reviewer is OK with keeping the ram semen data.

Reviewer #2 (Comments for the Author):

Kilama et al., in this manuscript, report the prevalence of *Fusobacterium necrophorum* (FNN, FNF) and *Fusobacterium varium* (FV) in reproductive microbiota of cattle and sheep, as well as in digestive tract ecosystems, using PCR and culture methods. Their findings reveal a high prevalence of FNF in bull semen (66.7%) and maternal ruminal fluids (87.1%). Importantly, their study is the first to suggest that *F. necrophorum* may be a commensal member of the healthy male reproductive microbiota.

Overall, this manuscript is well-written, with a robust experimental design and sound implementation. The research provides novel insights into the existence and distribution of *F. necrophorum* in cattle and sheep. The unexpected discovery of FNF being more prevalent than FNN and FV in cattle semen and ruminal fluids challenges previous findings that suggested FNN was the dominant species. The authors' explanation of these discrepancies is reasonable and highlights the complexity of studying *F. necrophorum*'s presence and pathogenicity.

Response: We appreciate the Reviewer's positive comments on our manuscript.

I only have two minor comments

1) P87-90: The sentences in this section are somewhat repetitive and could be revised for better clarity and conciseness. For instance:

"We and our group have previously reported that..."

Response: We have revised the statement to enhance clarity and conciseness (Lines 88-97).

2) The rationale for using qRT-PCR instead of conventional PCR to differentiate target genes for FNN, FNF, and FV requires clarification. Typically, qRT-PCR is employed to quantify gene expression levels, while conventional PCR, using specific primers, is more commonly used to identify specific strains based on the presence or size of the PCR product. Although qRT-PCR can potentially achieve similar results, it may not be as straightforward or practical for this purpose. Please elaborate on the decision to use qRT-PCR over conventional PCR. Additionally, providing a detailed description of the PCR protocols and primers would improve the manuscript's transparency and reproducibility.

Response: We appreciate the Reviewer's insightful comment. We have now provided a justification of our rationale for using qPCR instead of conventional PCR to differentiate target genes for FNN, FNF, and FV. Unlike conventional PCR, which primarily detects the presence or absence of a target sequence, qPCR offers higher sensitivity and specificity, allowing for the quantification of bacterial load in the samples. This quantitative capability was particularly useful for assessing relative abundance among different *Fusobacterium* species in our study. Additionally, qPCR reduces the likelihood of false positives by incorporating fluorescence-based detection, eliminating the need for post-PCR gel electrophoresis .

Furthermore, we have included a more description of the qPCR protocols, including annealing temperatures, reaction conditions, and controls, to enhance transparency and reproducibility. As cited, more details regarding the qPCR methods and primers can be obtained from the following article:

- A real-time PCR assay for the detection and quantification of *Fusobacterium necrophorum* and *Fusobacterium varium* in ruminal contents of cattle, Deters A, Xiaorong S, Jianfa B, Qing K, Jacques M, Nagaraja TG. 2024 Applied Animal Science 40:250-259, <https://doi.org/10.15232/aas.2023-02507>
- First report of isolation of *Fusobacterium varium* from liver abscesses and ruminal and colonic epithelial tissues of feedlot cattle, Applied Animal Science, Volume 40, Issue 3, 2024, Pages 244-249, ISSN 2590-2865, <https://doi.org/10.15232/aas.2023-02512>.

Reviewer #3 (Comments for the Author):

Summary: The manuscript from Kilama, et al describes a large-scale survey of the prevalence and viability of *Fusobacterium necrophorum* subsp. *necrophorum* and subsp. *funduliforme*, as well as *Fusobacterium varium* within bovine and ovine reproductive tracts. Some *Fusobacterium* species are traditionally thought of as livestock pathogens but previous work from this group suggests that some may also play a role in supporting fertility. Samples of bull and ram semen, bovine vaginal swabs and uterine tissue, as well as amniotic fluid and prenatal calf gut were screened via quantitative PCR for the presence of *Fusobacterium* using primer sets that distinguished between subsp. *necrophorum*, subsp. *funduliforme*, and *F. varium*. All three *Fusobacteria* were detected via this method, with bull semen and bovine maternal ruminal fluid having high prevalence of subsp. *funduliforme*. Paired samples were enriched for growth and plated onto media to confirm the presence of viable bacteria, with subsp. *funduliforme* being the overwhelming majority of detected species, again primarily in the bull semen and bovine maternal ruminal fluid. Species identity was confirmed via qPCR. Overall, this is a straightforward manuscript with assays that are well-performed, and the observations that *Fusobacterium necrophorum* subsp. *necrophorum* may be linked to fertility is therefore of economic interest for those in the livestock industry.

Specific comments:

1) How do the authors discriminate true seminal bacteria from those of the external bull urogenital tract? The semen was collected post-ejaculation, so it would have contacted any bacteria along the urethra. The authors even note that *Fusobacteria* are present in the bull prepuce in a previous study, so are the results in the current manuscript detecting seminal bacteria or urethral/prepuce bacteria? This seems to be an important distinction, even if the end result is the same, i.e., ejaculated semen from most bulls has subsp. *funduliforme* in it.

Response: We appreciate the reviewer's observation regarding the potential for contamination from the external urogenital tract during semen collection. To address this concern, we have added further details in the methods section about the steps taken to minimize external contamination. Specifically, semen samples were collected using sterile techniques, and the initial ejaculate fraction—more likely to contain contaminants from the urethra—was discarded prior to semen samples used for microbial analysis. Additionally, we compared control swabs (prepuce, room air, and collecting sheath) with semen samples to assess potential contamination (Lines 288-292).

While we acknowledge that some level of urethral/preputial microbial contribution is inevitable in post-ejaculatory semen samples, the consistent and predominant detection of *F. n.* subsp. *funduliforme* across multiple bull semen suggests that it may be a true seminal bacterium rather than an incidental contaminant. However, we recognize this as a limitation and acknowledge the need for further studies to develop less invasive pre-ejaculatory semen collection methods (e.g., epididymal aspiration) to confirm the true origin of these bacteria.

Furthermore, the seeding source and timing of colonization of the semen microbiota are not well documented which also warrants further investigation. We appreciate the reviewer's insightful comments and believe that these additions enhance the clarity and rigor of our study.

2) Do the authors have enough data to detect fertility differences between bulls and dams with viable *Fusobacteria* vs. those that were only detected via qPCR?

Response: We thank the Reviewer for raising this excellent question. While our current study provides valuable insights into the prevalence of *Fusobacterium* in bull semen and reproductive tract in female cattle, it was not specifically designed to assess fertility differences between individuals with viable *Fusobacterium* versus those detected only via qPCR. However, this study builds on our previous studies on bull semen and vagino-uterine microbiota characterization using the 16S rRNA amplicon sequencing. We reported a high relative abundance of genus *Fusobacterium* in bull semen (Webb et al., 2023a). We also observed greater relative abundance of *Fusobacterium* in uterine microbiota of beef cows that became pregnant to AI as compared to that of beef cows that remained open (Webb et al., 2023b), suggesting the positive association of *Fusobacterium necrophorum* with pregnancy. These observations from bull semen and uterine samples served as the basis for our hypothesis that the *Fusobacterium* spp. in bovine reproductive tract may have role in cattle fertility. Based on this hypothesis, we set out to explore the prevalence of the *Fusobacterium* spp. in male, female reproductive tract of healthy cattle. The future study should focus on establishing the correlation of viable *Fusobacterium* presence in bull reproductive tract and female reproductive tract with the male and female fertility.

The following are previous studies:

- Webb EM, Holman DB, Schmidt KN, Crouse MS, Dahlen CR, Cushman RA, Snider AP, McCarthy KL, Amat S. 2023. A Longitudinal Characterization of the Seminal Microbiota and Antibiotic Resistance in Yearling Beef Bulls Subjected to Different Rates of Gain. *Microbiol Spectr* 11:e05180-22. <https://doi.org/10.1128/spectrum.05180-22>

Webb EM, Holman DB, Schmidt KN, Pun B, Sedivec KK, Hurlbert JL, Bochantin KA, Ward AK, Dahlen CR, Amat S. Sequencing and culture-based characterization of the vaginal and uterine microbiota in beef cattle that became pregnant or remained open following artificial insemination. *Microbiol Spectr*. 2023 Dec 12;11(6):e0273223. doi: 10.1128/spectrum.02732-23. Epub 2023 Nov 3. PMID: 37921486; PMCID: PMC10714821.

3) The discussion section is the longest part of the manuscript by quite a margin and is somewhat repetitive. Removing redundancy and narrowing the focus would strengthen this section and improve readability.

Response: We thank the Reviewer's comments on the discussion. As suggested, we have revised the discussion of the manuscript with focus on the key findings to improve clarity and readability.

4) The text in the blue bar of table 3 has some words randomly between columns, so it is hard to judge which column they are supposed to be linked to.

Response: We thank the reviewer for pointing out this. We have revised the Table 3 by reformatting and color-blocking it based on the respective *Fusobacterium* species to provide clearer alignment and improved readability.

Re: Spectrum03145-24R1 (Characterizing the prevalence of *Fusobacterium necrophorum* subsp. *necrophorum*, *Fusobacterium necrophorum* subsp. *funduliforme* and *Fusobacterium varium* in bovine and ovine semen, bovine gut, vagino-uterine and fetal microbiota using targeted culturing and qPCR)

Dear Dr. Samat Amat:

Your manuscript has been accepted, and I am forwarding it to the ASM production staff for publication. Your paper will first be checked to make sure all elements meet the technical requirements. ASM staff will contact you if anything needs to be revised before copyediting and production can begin. Otherwise, you will be notified when your proofs are ready to be viewed.

Sincerely,
Artem Rogovsky
Editor
Microbiology Spectrum